# A syngeneic spontaneous zebrafish model of *tp53*-deficient, EGFR[vIII], and PI3KCA[H1047R]-driven glioblastoma reveals inhibitory roles for inflammation during tumor initiation and relapse in vivo

Alex Weiss[1], Cassandra D'Amata[1], Bret J Pearson[2,3,4]*, Madeline N Hayes[1,2]*

[1]Developmental and Stem Cell Biology Program, The Hospital for Sick Children, Toronto, Canada; [2]Department of Molecular Genetics, University of Toronto, Toronto, Canada; [3]Knight Cancer Institute, Oregon Health & Science University, Portland, United States; [4]Department of Pediatrics, Papé Research Institute, Oregon Health & Science University, Portland, United States

*For correspondence:
pearsobr@ohsu.edu (BJP);
madeline.hayes@sickkids.ca
(MNH)

Competing interest: The authors declare that no competing interests exist.

**Abstract** High-throughput vertebrate animal model systems for the study of patient-specific biology and new therapeutic approaches for aggressive brain tumors are currently lacking, and new approaches are urgently needed. Therefore, to build a patient-relevant in vivo model of human glioblastoma, we expressed common oncogenic variants including activated human EGFR[vIII] and PI3KCA[H1047R] under the control of the radial glial-specific promoter *her4.1* in syngeneic *tp53* loss-of-function mutant zebrafish. Robust tumor formation was observed prior to 45 days of life, and tumors had a gene expression signature similar to human glioblastoma of the mesenchymal subtype, with a strong inflammatory component. Within early stage tumor lesions, and in an in vivo and endogenous tumor microenvironment, we visualized infiltration of phagocytic cells, as well as internalization of tumor cells by *mpeg1.1*:EGFP+ microglia/macrophages, suggesting negative regulatory pressure by pro-inflammatory cell types on tumor growth at early stages of glioblastoma initiation. Furthermore, CRISPR/Cas9-mediated gene targeting of master inflammatory transcription factors *irf7* or *irf8* led to increased tumor formation in the primary context, while suppression of phagocyte activity led to enhanced tumor cell engraftment following transplantation into otherwise immune-competent zebrafish hosts. Altogether, we developed a genetically relevant model of aggressive human glioblastoma and harnessed the unique advantages of zebrafish including live imaging, high-throughput genetic and chemical manipulations to highlight important tumor-suppressive roles for the innate immune system on glioblastoma initiation, with important future opportunities for therapeutic discovery and optimizations.

## eLife assessment

This study presents a **valuable** syngeneic zebrafish model for studying glioblastoma and will be of interest to neuro-oncologists and cancer biologists. Using a feasible in vivo model to study the tumour microenvironment, cell/cell interaction, and immunity, the data are **compelling**, and opens up new lines of inquiries for future investigation on the impact of efferocytosis on tumor progression and cell of origin in this model as well as assessments of drug resistance mechanisms, using inhibitors to MAPK , Akt and/or mTOR pathway.

**eLife digest** Glioblastoma is the most common and deadly type of brain cancer in adults. Fewer than 7% of patients survive for more than five years after diagnosis. This poor prognosis for patients with glioblastoma has not significantly improved for decades.

The standard treatment for glioblastoma consists of surgery, radiotherapy and the same chemotherapy that has been prescribed for twenty years. This suggests that there is still much to learn about glioblastoma and how better to treat it. Scientists use various laboratory models to mimic human disease. They can study human glioblastoma cells grown in the laboratory or transplanted into mice, and they can also use genetically engineered mice that develop brain tumors from their own tissue.

These systems provide valuable information about glioblastoma, but each model has certain drawbacks. For example, glioblastoma cells in a dish do not grow in an environment containing other types of cells found in the body, such as immune cells. And although studying glioblastoma in mice bypasses this problem, such experiments often take years to perform and are very expensive.

To address these limitations, Weiss et al. asked whether introducing some of the same genetic mutations that cause glioblastoma in humans could lead to brain tumors in zebrafish. Zebrafish have multiple advantages as models of human disease: they are inexpensive to maintain and have a rapid life cycle, they are relatively easy to manipulate using various genetic tools, and they are transparent so that the growth of tumors can be filmed.

Weiss et al. expressed mutant versions of genes found in many patients with glioblastoma in the brains of developing zebrafish. These zebrafish rapidly developed tumor-like growths and detailed analyses confirmed that these tumors highly resembled human glioblastomas. Zebrafish glioblastomas contained active immune cells in addition to the cancer cells and showed signs of being inflamed.

Weiss et al. filmed interactions between immune cells and cancer cells in zebrafish brains. They noted that specific immune cells called macrophages (commonly known to destroy certain disease-causing pathogens like bacteria) had pieces of tumors inside them. This and other evidence suggested that these macrophages counteracted the growth of tumors by potentially engulfing (or 'eating') glioblastoma cells during the early stages of tumor development.

Altogether, these experiments indicate that zebrafish containing specific genes that cause glioblastoma in humans can mimic disease in many respects. Future studies will build on this work by testing other genes and further studying interactions between immune cells and cancer cells in the animal body.

## Introduction

Creating faithful models and discovering tailored treatments for patients with aggressive brain tumors has resulted in many different experimental platforms, each with their own unique advantages (*Gómez-Oliva et al., 2020*; *Haddad et al., 2021*). However, tractable vertebrate animal model systems for high-throughput study of tumor development in an intact and endogenous tumor microenvironment (TME) remain limited. New approaches that reflect patient genetics and physiology have the potential to aid in improving therapeutic strategies for individuals with malignant brain tumors like glioblastoma, which represents about 50% of all primary brain malignancies in adults (*Ostrom et al., 2017*; *Torp et al., 2022*).

Despite intensive treatments including surgery, radiation, and chemotherapy (temozolomide), most patients with glioblastoma eventually relapse and have a median survival rate of less than 15 months (*Aldape et al., 2019*). Therapeutic resistance can partly be attributed to a poor understanding of underlying molecular mechanisms, as well as a significant level of heterogeneity between patients and within individual tumors. Molecular heterogeneity has been important for the classification of three major subtypes of glioblastoma including proneural, classical, and mesenchymal (*Brennan et al., 2013*; *Torp et al., 2022*; *Verhaak and Valk, 2010*; *Wang et al., 2017*). However, recent evidence supporting co-existence of inter-converting glioblastoma cell states within individual patient tumors reveals less distinct subtype separations and phenotypic flexibilities that contribute to the aggressiveness and drug resistance of glioblastoma across molecular cohorts (*Darmanis et al., 2017*; *Neftel et al., 2019*; *Patel et al., 2014*; *Couturier et al., 2020*).

In general, 90% of glioblastoma tumors display alterations in core signaling factors involved in receptor tyrosine kinase (RTK)/RAS/PI3K pathway signaling (*Brennan et al., 2013*; *McLendon et al., 2008*). While activating mutations in RAS proteins are rarely found in the clinic, loss of the negative regulator *NF1* is common, as well as amplifications and/or activating mutations in various RAS pathway proteins including epidermal growth factors receptor (EGFR), which are found in >50% of glioblastomas (*Brennan et al., 2013*; *Hoogstrate et al., 2022*; *McLendon et al., 2008*). Loss of the lipid phosphatase PTEN and/or activating mutations in the PI3K catalytic subunit PIK3CA are also commonly found, which altogether drive downstream activation of oncogenic RAS/MAPK and AKT/mTOR signaling, among other crucial pathways involved in growth and survival of glioblastoma (*Brennan et al., 2013*; *McLendon et al., 2008*). Furthermore, the TP53 tumor suppressor pathway is altered in 84% of globlastoma patients and 94% of cell lines, with TP53 loss implicated in tumor cell proliferation, invasion, migration, and stemness (*Brennan et al., 2013*; *McLendon et al., 2008*; *Zhang et al., 2018*). Importantly, single pathway mutations are generally insufficient to transform normal brain tissues, and multiple mutations are required for glioma formation (*Chen et al., 2018*; *Haddad et al., 2021*; *Holland et al., 2013*; *Takke et al., 1999*). Therefore, to recapitulate human glioblastoma in experimental models, multiple genetic events should be considered.

In addition to key genetic drivers, the TME is known to influence cellular flexibility in glioblastoma and consists of heterogeneous collections of resident brain, stroma, and immune cells, as well as cells recruited from the general circulation such as bone marrow-derived immune cells with known tumor-suppressive and tumor-promoting functions (*Bikfalvi et al., 2023*; *Quail and Joyce, 2017*). Given emerging opportunities for harnessing the immune system for the treatment of human cancer, there is a growing focus on understanding innate and adaptive immune responses across different subtypes of human malignancies. However, like many other tumors, glioblastoma combines a lack of immunogenicity due to few mutations with a highly immunosuppressive tumor microenvironment (TME). In addition to off-target effects of current frontline therapeutic strategies, both tumor and immune cells contribute to immune suppression in and surrounding the TME in glioblastoma (*McGranahan et al., 2019*; *Sengupta et al., 2012*), which could explain the failure of immunotherapy-based clinical trials. Additionally, lymphocytes are frequently exhausted and dysfunctional and therefore inadequate at exerting an anti-tumor immune response, while tumor-associated myeloid cells are frequently reprogrammed by signaling from tumor cells and the TME to cell states that promote glioblastoma survival, growth, and invasion (*Kennedy et al., 2013*; *McGranahan et al., 2019*; *Pearson et al., 2020*). Evidence also supports tumor cell-intrinsic mechanisms in response to immune cell attack leading to various evasion mechanisms, including upregulation of myeloid-associated gene expression programs and resistance to interferon signaling (*Gangoso et al., 2021*; *Parmigiani et al., 2022*).

Complex intercellular communication in glioblastoma highlights the importance of faithful in vivo models. However, roles for the endogenous TME especially at early stages of tumor initiation, remain poorly understood. Furthermore, live visualization of heterogeneity and tumor cells within the TME is limited in non-transparent rodent genetic models and/or patient-derived tumor xenografted hosts. Therefore, here our goal was to develop a patient-relevant model of aggressive human glioma in an intact and immune-competent system. We developed a novel spontaneous, syngeneic zebrafish model of glioblastoma, with high levels of inflammatory immune cell infiltration and anti-tumor associations between phagocytes in the TME and tumor cells, suggesting inhibitory roles for intercellular interactions during glioblastoma initiation and an effective in vivo platform for future biological discovery and drug testing for patients.

## Results

### Oncogenic MAPK/AKT pathway activation drives glial-derived tumor formation in syngeneic *tp53* mutant zebrafish

To generate a patient-relevant brain tumor model in zebrafish, we used the zebrafish *her4.1* promoter to simultaneously over-express constitutively active human EGFR (EGFR$^{vIII}$) and PI3KCA (PI3KCA$^{H1047R}$) variants in neural progenitors and radial glia of syngeneic *tp53* loss-of-function mutant larvae (*Ignatius et al., 2018*; *Takke et al., 1999*; *Than-Trong et al., 2020*). Co-injection of linearized *her4.1*:**E**GFR$^{vIII}$ + *her4.1*:**P**I3KCA$^{H1047R}$ + *her4.1*:m**S**carlet transgenes into syngeneic (CG1) *tp53$^{-/-}$* mutant embryos at the one-cell stage led to broad transient mScarlet expression for 5–6 days followed by rare mosaic

expression in the anterior CNS (henceforth referred to as p53EPS, *Figure 1A*), as expected from a transient mosaic injection strategy used to express stable concatemers of DNA vectors (*Langenau et al., 2008*). At 15 days post fertilization (dpf), this transgene combination led to visible mScarlet-positive brain lesions in the anterior CNS of live zebrafish (*Figure 1B–D*). Injections of single linearized vector and vector combinations resulted in a maximum incidence of approximately 15–20% affected zebrafish induced by the p53EPS combination, by 45 dpf (*Figure 1D*). Intertumoral variability was observed among p53EPS mosaic-injected zebrafish, with tumors of variable size arising in different brain regions including the telencephalon and diencephalon regions (8/29=27.5%, from three independent tumor screens), and the optic tectum/mesencephalon region (21/29=72.5%, from three independent tumor screens) (*Figure 1—figure supplement 1*). To further define the tissue of origin of zebrafish p53EPS CNS lesions, we co-injected linearized *gfap*:EGFP (*Don et al., 2017*) and visualized EGFP expression in 100% of brain masses at 30–40 dpf (*Figure 1A and B*), supporting glial identity and a novel in vivo model of malignant glioma.

To test for transformation of p53EPS cells from primary mosaic-injected animals, we harvested bulk tissue from dissected zebrafish brains and transplanted dissociated cells into the hindbrain ventricles of 2 dpf syngeneic *tp53* wild-type (CG1) zebrafish (*Mizgireuv and Revskoy, 2006*). At approximately 18 days post-transplant (18 dpt, or 20 dpf), we screened for mScarlet+ fluorescence and visualized her4.1+/gfap+ tumor cell outgrowth in 16–25% of transplanted hosts (n=3 independent screens, *Figure 1E and F*), supporting oncogenic transformation and malignant growth in vivo of p53EPS cells. Fluorescence-activated cell sorting (FACS) revealed fluorescently labeled tumor cells in un-labeled syngeneic host brains and co-expression of *her4.1*:mScarlet/*gfap*:EGFP (*Figure 1G*), further supporting tumor growth from a glial-derived progenitor cell, which was expected from *her4.1*-specific oncogene activation during zebrafish development (*Than-Trong et al., 2020*).

We performed serial sectioning of uninjected control and primary p53EPS zebrafish brains and performed histological characterizations, which revealed highly aggressive and proliferative tumors in p53EPS consisting of a heterogenous mix of different malignant cell types, compared to normal brains (*Figure 1H–O*, *Figure 1—figure supplement 2*). One major cell type displayed a round granular-like morphology with a high nuclear to cytoplasm ratio, while a second cell type found within the same tumor masses displayed a flattened and elongated morphology (*Figure 1L*, *Figure 1—figure supplement 2*). In general, tumors displayed clean margins without invasive boundaries, necrosis, or microvascular proliferation; however, a few instances of cell migration along vasculature were detected. Certain p53EPS tumors displayed embryonal features including rosette-type structures and a high mitotic index, overlapping histological features with glioblastoma with primitive neuronal component (*Suwala et al., 2021*), which may be expected given our chosen combination of molecular drivers as well as a neural progenitor and/or radial glial cell of origin, during developmental stages.

To assess oncogenic signaling pathway activation, we stained unaffected and p53EPS tumor-positive brain sections for phosphorylated-ERK (p-ERK) and phosphorylated-AKT (p-AKT), indicators of activated MAPK and AKT signaling pathways, respectively (*Figure 1J–K and N–O*). Compared to normal brains, we found increased p-ERK and p-AKT staining within tumor lesions, consistent with elevated MAPK and AKT activation driving malignant transformation (*Figure 1N–O*). To further validate effects of relevant downstream signaling pathway activation, we also co-injected linearized *her4.1*:**K**RAS$^{G12D}$ + *her4.1*:**P**I3KCA$^{H1047R}$ + *her4.1*:**E**GFP into *tp53*$^{-/-}$ embryos (p53KPG). At 15–20 dpf, we visualized tumor onset and penetrance comparable to p53EPS (*Figure 1—figure supplement 3*), suggesting a dominant role for MAPK/AKT pathway activation downstream of RTK signaling in driving tumor formation, and a flexible oncogenic strategy for inducing robust brain tumor formation in zebrafish.

## Gene set enrichment analyses reveal gene expression patterns consistent with human mesenchymal glioblastoma and inflammation

To further characterize our novel zebrafish brain tumor models, we performed bulk RNA sequencing (RNAseq) of three independent p53EPS tumor-burdened brains, three independent p53KPS tumor-burdened brains, and three age-matched, control-injected zebrafish brains that remained tumor-free at the time of harvesting tissue. Using hierarchical clustering on normalized gene expression, our tumor-free control samples clustered together and apart from p53EPS and p53KPS tumor brains, which displayed a recognizable level of transcriptional variability across principal components

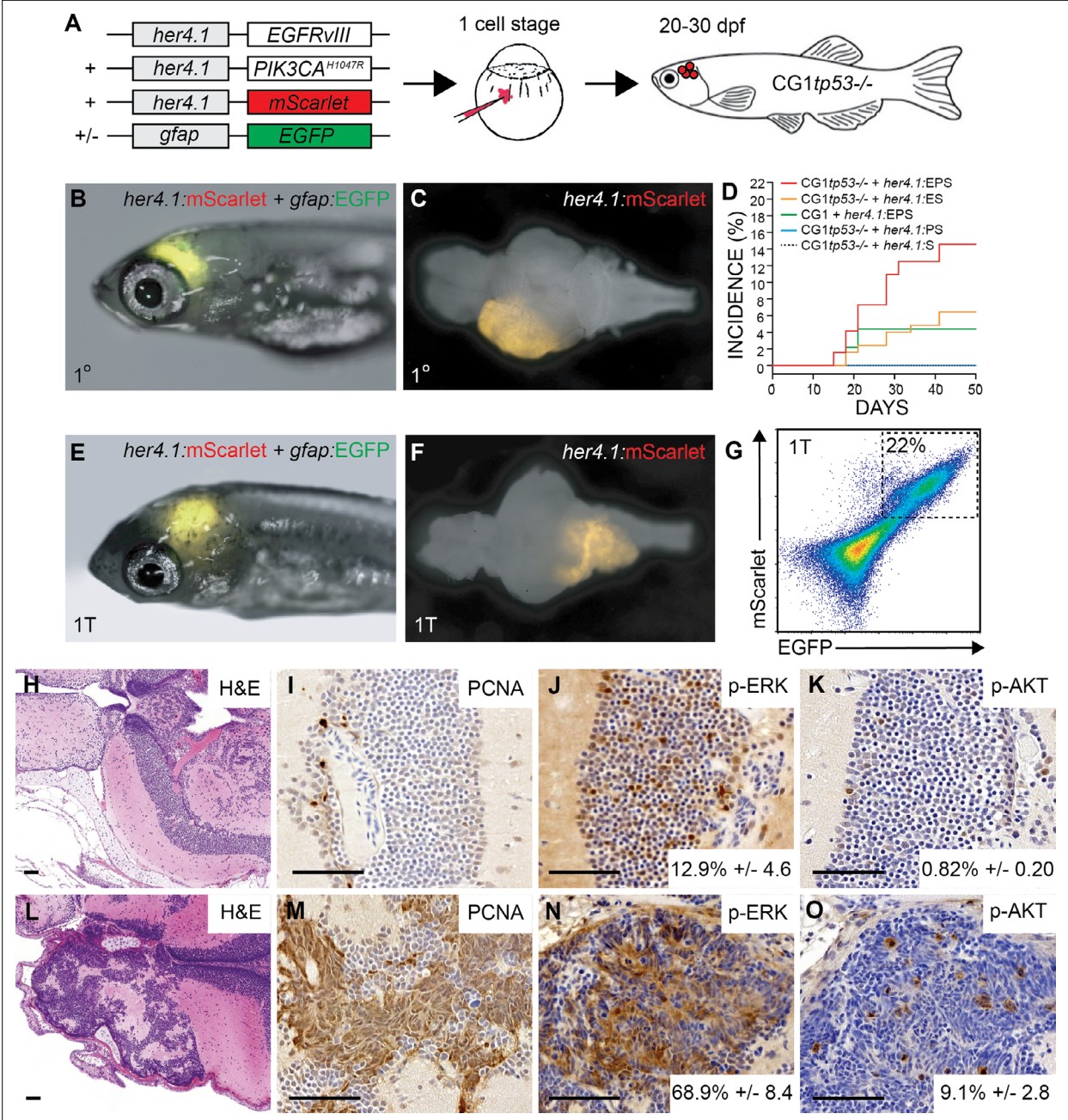

**Figure 1.** *her4.1*-driven over-expression of patient-relevant oncogenes drives glial-derived brain tumor formation in syngeneic *tp53* loss-of-function mutant zebrafish. (**A**) Schematic of modeling strategy where linearized transgene vectors with the zebrafish *her4.1* promoter driving human EGFR[vIII], human PI3KCA[H1047R], and mScarlet fluorescent proteins are co-injected at the one-cell stage into syngeneic (CG1 strain) *tp53*[-/-] mutant zebrafish embryos. Starting at 15 days post fertilization (dpf), mosaic-injected zebrafish were screened for CNS tumor formation, indicated by mScarlet expression in the brain region of live zebrafish. Co-injection of *gfap*:GFP linearized transgene is used to assess glial-specific cell fate specification in vivo. (**B**) *her4.1*:mScarlet and *gfap*:GFP expression in the anterior CNS of mosaic-injected syngeneic (CG1 strain) *tp53*[-/-] zebrafish at 30 dpf. (**C**) Whole brain dissected from a p53EPS mosaic-injected zebrafish at 30 dpf. (**D**) Cumulative frequencies of mScarlet+ CNS lesions in syngeneic *tp53*[-/-] mutant (CG1*tp53*[-/-]) and wild-type (CG1) zebrafish injected at the one-cell stage with *her4.1*:EGFR[vIII] (**E**), *her4.1*:PI3KCA[H1047R] (**P**), and/or *her4.1*:mScarlet (**S**). (**E**) Syngeneic (CG1 strain) zebrafish at 30 dpf engrafted with *her4.1*:mScarlet+/*gfap*:GFP+ brain tumor cells, following primary transplantation (1T) at 2 dpf into the embryonic brain ventricle. (**F**) Whole brain dissected from engrafted syngeneic host (CG1) zebrafish at 30 dpf. (**G**) Fluorescence-activated cell sorting (FACS) plot of bulk syngeneic host brain following primary transplant (1T) of *her4.1*:EGFR[vIII] + *her4.1*:PI3KCA[H1047R] + *her4.1*:mScarlet + *gfap*:GFP brain tumor cells. (**H–O**) Histological staining of uninjected control (**H–K**) and p53EPS tumor-burdened brains (**L–O**). (**H, L**) Hematoxylin and eosin (H&E) staining of coronal sections highlighting telecephalon and diencephalon regions of representative control (**H**) and p53EPS (**L**) brains. (**I, M**) Proliferating cell nuclear antigen (PCNA) staining of control (**I**) and p53EPS (**M**) brain sections. (**J, N**) Phosphorylated-ERK (p-ERK) staining and quantifications reveal

*Figure 1 continued on next page*

*Figure 1 continued*

increased MAPK signaling pathway activation in p53EPS tumors (p<0.001, n=3 independent tumor sections). (**K, O**) Phosphorylated-Akt (p-Akt) staining and quantifications reveal increased Akt signaling pathway activity in p53EPS tumors (p=0.007, n=3 independent tumor sections). Scare bars represent 50 µm.

The online version of this article includes the following figure supplement(s) for figure 1:

**Figure supplement 1.** Intertumoral heterogeneity in p53EPS-induced tumors.

**Figure supplement 2.** Hematoxylin and eosin (H&E) staining of three independent p53EPS tumors.

**Figure supplement 3.** *her4.1*-driven over-expression of KRAS$^{G12D}$ + PI3KCA$^{H1047R}$ drives glial-derived brain tumor formation in syngeneic *tp53* loss-of-function mutant zebrafish.

(*Figure 2A*). Interestingly, p53EPS and p53KPS samples failed to cluster according to oncogenic drivers (*Figure 2A*), suggesting molecular similarities as well as inter-tumor heterogeneity reflecting differences in tumor location, size, and contribution of tumor cells to total sample inputs.

Given the relevance of our p53EPS driver combination to human glioblastoma, we chose to focus the remainder of our molecular analyses on p53EPS samples. Using differential gene expression analysis, we identified a conserved set of differentially expressed (DE) genes in p53EPS tumor brains, with 236 significantly upregulated and 28 downregulated genes, compared to control-injected brains at 20–30 dpf (2>log2foldChange>–2, adjusted p-value<0.05, *Supplementary file 1*, *Supplementary file 2*, *Figure 2B*). Using human orthologs of DE genes, we performed gene set enrichment analysis (GSEA) (*Mootha et al., 2003*; *Subramanian et al., 2005*), comparing our p53EPS zebrafish model to published expression patterns for human glioblastoma subtypes, as well as embryonal brain tumors including designated subtypes of medulloblastoma (*Cavalli et al., 2017*; *McLendon et al., 2008*; *Wang et al., 2017*). Among these gene sets, we found a significant enrichment for the mesenchymal subtype of human glioblastoma in p53EPS DE genes (*Figure 2C*, *Supplementary file 3*; *McLendon et al., 2008*; *Wang et al., 2017*). In contrast, while certain genes for other glioblastoma and/or medulloblastoma subtypes were upregulated in p53EPS samples, no significant enrichment was found for these specific gene expression signatures (*Figure 2—figure supplement 1*, *Supplementary file 3*; *Cavalli et al., 2017*; *McLendon et al., 2008*; *Wang et al., 2017*).

To assess potential underlying molecular mechanisms involved in p53EPS formation, we assessed enrichment for Hallmark gene expression sets available through the Molecular Signatures Database (MSigDB, *Supplementary file 4*; *Villanueva et al., 2011*). Interestingly, 7 of the top 13 enriched gene sets identified (NOM p-value<0.05) related to inflammation or inflammatory signaling pathways, including the interferon gamma response, TNFA signaling, the interferon alpha response, and Jak/STAT3 signaling (*Figure 2D*, *Supplementary file 4*), suggesting a strong inflammatory component in our brain tumor model. Additional pathway signatures included those related to RAS signaling, hypoxia, and epithelial-to-mesenchymal transitions (*Supplementary file 4*), suggesting on-target oncogenic pathway activation, hypoxia, and invasive properties, consistent with aggressive glioblastoma (*Majc et al., 2020*; *Park and Lee, 2022*). Therefore, bulk RNAseq expression data supports significant molecular similarity between our zebrafish p53EPS brain tumor model and human glioblastoma of the mesenchymal subtype, with a significant inflammatory component.

Given the in vivo context and contribution of tumor and non-tumor cell types to our bulk RNAseq analysis, we decided to assess RNA expression in FACS-sorted *her4.1*:mScarlet-positive tumor cells, as well as *her4.1*:mScarlet-negative bulk stromal cells, compared to non-tumor whole brain tissue (*Figure 2E*, *Supplementary file 5*). Interestingly, we observed increased immune cell and inflammatory gene expression in both p53EPS tumor and stromal cell fractions compared to control unaffected whole brain tissue, including transcripts associated with myeloid cell types (*mpeg1, irf7, irf8*), lymphoid progenitor and specific subtype regulators (*rag1, rag2, lck*), *stat1a/b*, *fas* cell surface death receptor, and Toll-like receptor 4b (*tlr4bb*), among other genes involved in immune responses (adjusted p-value<0.0001, *Figure 2E*, *Supplementary file 5*). We validated a selection of genes from RNAseq using RT-PCR (*Figure 2—figure supplement 2*), as well as transient co-injection of linearized transgene with our EPS mix to assess tumor cell-specific expression of lymphoid *rag2*:EGFP and myeloid *mpeg1.1*:EGFP at 30 dpf (*Figure 2F–I*). Interestingly, we observed co-expression of *her4.1*:mScarlet/*rag2*:EGFP and *her4.1*:mScarlet/*mpeg1.1*:EGFP in p53EPS tumors (*Figure 2F–I*), suggesting promoter activity at certain immune-associated genes within established p53EPS tumor cells in vivo.

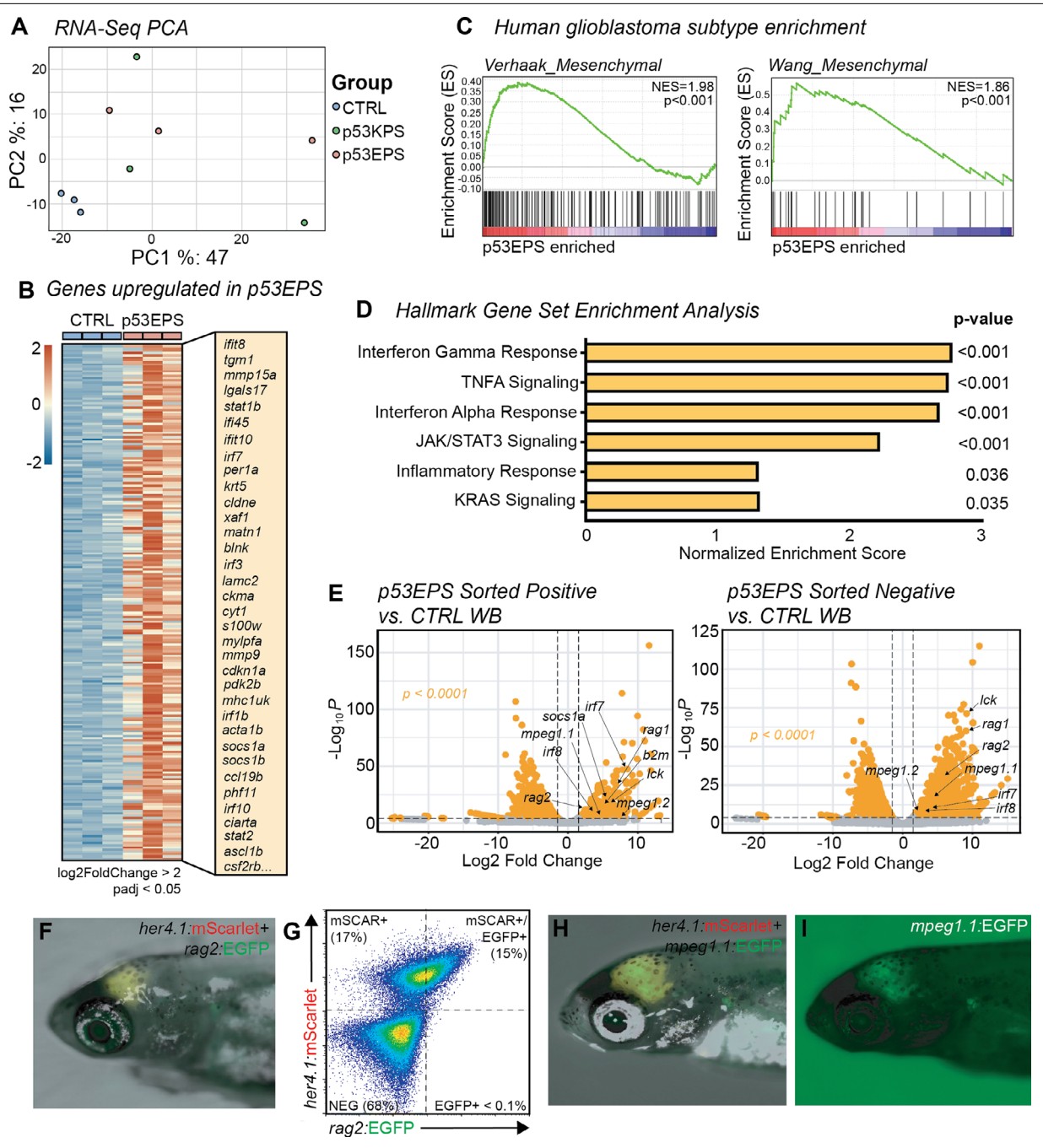

**Figure 2.** RNA expression analysis establishes enrichment of mesenchymal glioblastoma and inflammation signatures in p53EPS model. (**A**) Principal component analysis (PCA) of mRNA sequencing from whole control-injected brains (CTRL), p53EPS, and p53KPS tumor-burdened brains. (**B**) Heatmap of normalized counts for genes upregulated in p53EPS tumor-burdened brains (log2foldChange>2, padj<0.05), compared to whole control-injected brains (CTRL). A selected list of upregulated transcripts is indicated. (**C**) Gene set enrichment analysis (GSEA) plots of published gene signatures for mesenchymal subtype glioblastoma for genes differentially regulated in p53EPS compared to control-injected brains (*McLendon et al., 2008*; *Wang et al., 2017*). Normalized enrichment scores (NES) and nominal p-values are indicated. (**D**) Bar plot of NES from GSEA of Hallmark gene sets (*Villanueva et al., 2011*). (**E**) Volcano plots of differentially expressed genes between sorted mScarlet+ p53EPS tumor cells and control-injected whole brain tissue (CTRL WB), as well as between sorted mScarlet-negative cells from p53EPS tumor-burdened brains and control-injected whole brains (CTRL WB). (**F**) *her4.1*:mScarlet and *rag2*:EGFP expression in live zebrafish with a p53EPS tumor at 30 days post fertilization (dpf). (**G**) Fluorescence-activated cell sorting (FACS) plot of p53EPS brain with *rag2*:EGFP co-expression from (**F**). (**H, I**) *her4.1*:mScarlet and *mpeg1.1*:EGFP expression in live zebrafish with a p53EPS tumor at 30 dpf.

The online version of this article includes the following figure supplement(s) for figure 2:

*Figure 2 continued on next page*

*Figure 2 continued*

**Figure supplement 1.** Gene set enrichment analysis (GSEA) plots of published gene signatures for alternative molecular subtypes of human glioblastoma and medulloblastoma for genes differentially regulated in p53EPS compared to control-injected brains (*Cavalli et al., 2017*; *McLendon et al., 2008*; *Wang et al., 2017*).

**Figure supplement 2.** Quantitative real-time PCR analysis of neural stem cell (NSC) genes and genes associated with inflammatory gene expression signatures identified using bulk RNA sequencing.

These data suggest inflammation-associated gene expression in both zebrafish glioblastoma-like cells and the TME and is consistent with inflammatory gene expression in tumor cells associated with immune evasion and in vivo growth in other models (*Gangoso et al., 2021*; *Yang et al., 2022*). Altogether, our in vivo expression data supports p53EPS zebrafish as a comparable and relevant model system to study glioblastoma tumor biology, as well as intercellular interactions within an endogenous TME.

## Zebrafish mesenchymal glioblastoma-like tumors recruit activated mononuclear phagocytes at early stages of tumor formation

Given that p53EPS tumor-burdened brains displayed strong enrichment for transcriptional signatures associated with inflammatory signaling, we were interested in the role for innate immune cells in the earliest stages of p53EPS tumor formation. Myeloid cells including microglia and macrophages are recruited in many different subtypes of primary brain tumors and in brain metastases (*Gutmann and Kettenmann, 2019*; *Khan et al., 2023*). However, how phagocyte populations affect glioblastoma initiation is less well understood given that most studies utilize established tumor cell models and/or patient-derived tissue xenograft transplantations into immune-compromised host animals. To first determine whether myeloid-derived phagocytic cell lineages are enriched in p53EPS lesions at early stages of tumor initiation, we used neutral red staining that labels lysosomal-rich phagocytes, irrespective of cell type (*Figure 3A–C*; *Herbomel et al., 2001*; *Shiau et al., 2015*). At 10 dpf, we observed enrichment of neutral red-positive foci in regions of *her4.1*:mScarlet+ fluorescent intensity (*Figure 3B and C*), suggesting phagocyte infiltration during p53EPS initiation, prior to typical observation of macroscopic tumor masses (*Figure 1D*). To assess the activation state of phagocytes in and surrounding p53EPS tumors, we co-injected our oncogene combination into $tp53^{-/-}$ embryos carrying a Tg(*tnfa*:EGFP) transgenic marker of activated and pro-inflammatory phagocytes (*Hao et al., 2012*; *Nguyen-Chi et al., 2015*), and observed enrichment of *tnfa*:EGFP single-positive cells in *her4.1*:mScarlet+ lesions (*Figure 3D and E*). We also performed GSEA against immunologic signature gene sets from the MSigDB (*Villanueva et al., 2011*; *Mootha et al., 2003*; *Subramanian et al., 2005*), and identified multiple enriched gene expression signatures associated with inflammation and inflammatory cell types in p53EPS tumor-burdened brains (*Supplementary file 6*). p53EPS displayed features of both pro- and anti-inflammatory gene expression, which is consistent with a mixed immune activation/ suppressive state in glioblastoma known to play an important role in tumor progression in patients (*Biswas et al., 2008*; *Karimi et al., 2023*; *Quail and Joyce, 2017*; *Ren et al., 2023*). However, p53EPS tissues more significantly enriched for gene expression associated with activated and pro-inflammatory phagocytes including classical M1 macrophages, compared to alternative M2 polarized macrophages, and genes typically downregulated during M2 polarization following macrophage colony stimulating factor treatment (*Figure 3—figure supplement 1*, *Supplementary file 6*), further suggesting strong promotion of inflammation within the TME.

Given enrichment for *tnfa*:EGFP+ phagocytes and signatures associated with pro-inflammatory macrophages in our glioblastoma model, we decided to investigate microglia/macrophages cell dynamics in vivo using live confocal imaging following co-injection of the linearized transgene combination into $tp53^{-/-}$; Tg(*mpeg1.1*:EGFP) zebrafish (*Ellett et al., 2011*). We first looked at a time-course of lesion formation at 5, 7, and 9 dpf relative to microglia in the zebrafish brain (*Figure 3F–K*), which colonize the zebrafish neural retina by 48 hpf, and the optic tectum by 84 hpf (*Herbomel et al., 2001*). At 5 dpf (120 hpf), we observed comparable levels of *mpeg1.1*:EGFP+ microglia throughout the cephalic region and in the brain in both control and p53EPS-injected zebrafish (*Figure 3F, I*). p53EPS-injected zebrafish brains displayed mosaic cellular expression of *her4.1*:mScarlet; however, cells were sparse and diffuse, and no large intensely fluorescent tumor-like clusters were detected at this stage (*Figure 3I*, n=12/12 tumor negative). At 7 dpf, we detected

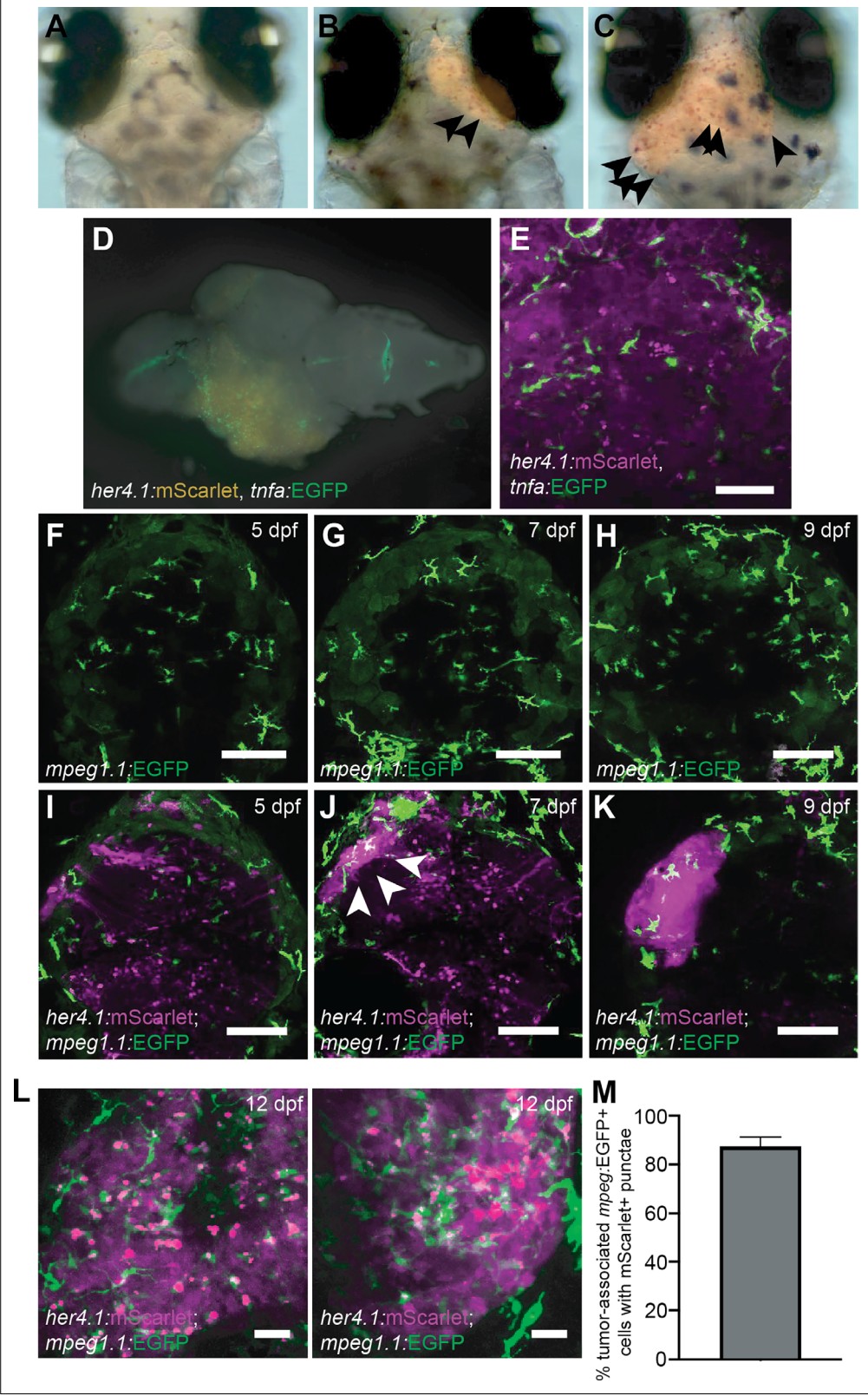

**Figure 3.** p53EPS recruits activated microglia/macrophages at early stages of tumor initiation. (**A–C**) Neutral red staining of p53EPS mScarlet tumor-negative (**A**) and mScarlet tumor-positive brains (**B, C**) at 10 days post fertilization (dpf). Neutral red foci in early-stage lesions are highlighted with arrows and are indicative of phagocytic cells. (**D**) Whole brain with p53EPS-induced tumor in a transgenic Tg(*tnfa*:EGFP) zebrafish at 20 dpf.

*Figure 3 continued on next page*

*Figure 3 continued*

(**E**) Z-stack projection of live confocal imaging of p53EPS tumor in transgenic Tg(*tnfa*:EGFP) background. (**F–K**) Z-stack projections of control uninjected (**F–H**) and p53EPS brains (**I–K**) at 5 dpf (**F ,I**), 7 dpf (**G, J**), and 9 dpf (**H, K**) in transgenic Tg(*mpeg1.1*:EGFP) background. White arrows highlight an early-stage p53EPS lesion and associated *mpeg*:EGFP+ cells. (**L**) Z-stack projections of two independent p53EPS brains at 12 dpf in transgenic Tg(*mpeg1.1*:EGFP) background. (**M**) Quantification of tumor-associated *mpeg1.1*:EGFP+ cells with overlapping and/or internalized *her4.1*:mScarlet+ punctae (n=3 independent tumors).

The online version of this article includes the following figure supplement(s) for figure 3:

**Figure supplement 1.** Gene set enrichment analysis (GSEA) plots of established gene signatures for classical M1 polarized macrophages (Classical_M1_VS_Alternative_M2_Macrophage_UP), compared to alternative M2 macrophages (Classical_M1_VS_Alternative_M2_Macrophage_DN).

small clusters of *her4.1*:mScarlet+ cells (***Figure 3J***, n=5/9), indicative of early lesion formation. Interestingly, these lesions were surrounded and/or infiltrated by *mpeg1.1*:EGFP+ microglia, while at 9 dpf *her4.1*:mScarlet+ expression became highly specific to tumor lesions, and these lesions were consistently associated with *mpeg1.1*:EGFP+ microglia/macrophages (***Figure 3K***, n=8/8 tumor-positive zebrafish).

At 12 dpf in p53EPS zebrafish with established tumors, we observed fluorescent *mpeg1.1*:EGFP+ cells surrounding and within regions of concentrated *her4.1*:mScarlet+ fluorescence (***Figure 3L***, ***Video 1***). Microglia/macrophages outside of early-stage *her4.1*:mScarlet+ lesions displayed highly ramified morphologies, with several processes that were extended and retracted, indicative of environmental surveillance (***Figure 3L***, ***Video 1***; ***Nimmerjahn et al., 2005***). *mpeg1.1*:EGFP+ microglia/macrophages infiltrated into dense *her4.1*:mScarlet+ regions displayed more rounded and amoeboid-like morphology, supporting their activation in association with p53EPS oncogenic cells (***Karperien et al., 2013***). Interestingly, *mpeg1.1*:EGFP+ microglia/macrophages dynamically interacted with *her4.1*:mScarlet+ cells (***Video 1***, ***Video 2***), and in p53EPS oncogenic masses, *mpeg1.1*:EGFP+ microglia/macrophages associated closely with and often displayed internalized *her4.1*:mScarlet+ punctate cells (***Figure 3L and M***), suggesting engulfment and removal of p53EPS cells during tumor formation in vivo. Together with our expression data, visualization of infiltrating myeloid-derived immune cells in p53EPS glioblastoma-like lesions, dynamic *mpeg*+ microglia/macrophages-p53EPS interactions, and tumor cell engulfment suggests anti-tumoral activity at early stages that could negatively affect tumor formation in vivo.

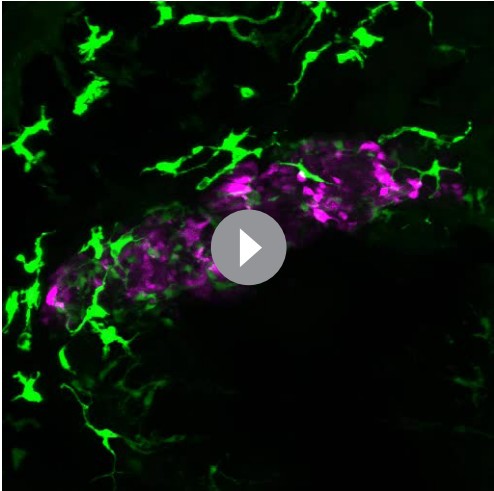

**Video 1.** Time-lapse confocal images of p53EPS brain at 12 days post fertilization (dpf) in transgenic Tg(*mpeg1.1*:EGFP) background.

https://elifesciences.org/articles/93077/figures#video1

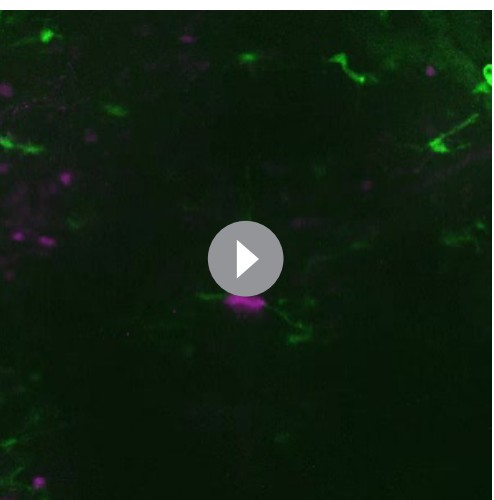

**Video 2.** Time-lapse confocal images of individual *her4.1*:mScarlet+ p53EPS and *mpeg1.1*:EGFP+ cells at 12 days post fertilization (dpf).

https://elifesciences.org/articles/93077/figures#video2

## Inflammation-associated *irf7* and *irf8* are required to inhibit p53EPS tumor formation in vivo

Interferon regulatory factor (Irf) proteins regulate transcription of interferon genes and support a variety of different immune reposes. Irf7 and Irf8 are critical for global activation of the type I IFN response following stimulation and for myeloid cell development, respectively (*Günthner and Anders, 2013*; *Ning et al., 2011*; *Shiau et al., 2015*). Irf8 is a conserved determinant of macrophage cell fate, as well as dendritic cell survival and function, and *irf8* mutant zebrafish lack microglia/macrophages in the brain up to 31 dpf (*Shiau et al., 2015*; *Sichien et al., 2016*). Irf7 also drives differentiation of macrophages; however, in zebrafish and other systems, Irf7 is more broadly activated in immune cells in response to infection, with evidence also supporting non-immune cell-related functions during development and cancer progression (*Feng et al., 2016*; *Gangoso et al., 2021*; *Günthner and Anders, 2013*; *Hu et al., 2022*; *Ning et al., 2011*; *Yang et al., 2022*). Interestingly, we observed upregulation of several interferon regulatory factor (Irf) family members including *irf7* and *irf8* in p53EPS tumor-burdened brains (*Figure 2B*, *Supplementary file 1*, *Supplementary file 2*), suggesting potential roles for these master regulators of inflammation on immune-related responses during p53EPS tumorigenesis in vivo. Therefore, to assess functional roles for Irf7 and Irf8 in p53EPS formation, we used a transient CRISPR/Cas9 gene targeting approach to knock down *irf7* or *irf8* genes prior to p53EPS tumor formation using co-injection of two to three guide RNAs (gRNAs) each targeting *irf7* or *irf8* (*Supplementary file 7*), together with Cas9 protein and linearized EPS into one-cell stage *tp53⁻ᐟ⁻* embryos. At 2–5 dpf, we extracted DNA from a subset of injected embryos and observed a gene targeting efficiency of >90% and >65% INDELS at the *irf7* or *irf8* loci, respectively (ICE Analysis, https://ice.synthego.com/#/, *Synthego, 2019*). Gene transcript knock-down was also verified using RT-PCR (*Figure 4—figure supplement 1*). Consistent with previous reports, *irf8* knock-down resulted in significant reductions in neutral red-positive phagocytes at 8–10 dpf (*Figure 4—figure supplement 2*; *Shiau et al., 2015*), and while no significant differences in neutral red-positive phagocyte number were observed following *irf7* gene targeting, *irf7* CRISPR/Cas9-injected animals displayed early mortality un-related to brain tumor formation, with ~80% of CRISPR-injected animals displaying illness prior to 2 months of age, consistent with a broad requirement for Irf7 in immune responses, among other functions.

Remarkably, following *irf7* or *irf8* gene knock-down with p53EPS, we observed robust tumor formation, with 65% of *irf7*-targeted and 42% of *irf8*-targeted p53EPS zebrafish developing tumors by 30 dpf, compared to 20% p53EPS incidence (*Figure 4A–D*, *irf7* p<0.0001, *irf8* p=0.0155, Fisher's exact test), suggesting an important inhibitory role for *irf7* and *irf8* in p53EPS tumor initiation. In p53EPS tumor brains with *irf7* and *irf8* knock-down, we observed reduced expression of genes associated with our inflammation signature in p53EPS tumor-burdened brains, including immune evasion-associated transcripts like *suppressor of cytokine signaling 1a* (*socs1a*) (*Figure 4—figure supplement 1*), suggesting reduced tumor-specific inflammation and associated immune evasion mechanisms, which were previously shown to be upregulated in response to anti-tumor cell infiltration (*Gangoso et al., 2021*). To further validate gene targeting and assess the spatial localization of *irf7* expression within the TME, we performed fluorescent in situ hybridization (FISH) in control and *irf7* knock-down tumors and detected robust transcript expression within early-stage p53EPS lesions at 8 dpf (*Figure 4E*). In animals co-injected with *irf7* CRISPR, we detected p53EPS tumor formation, albeit with reduced *irf7* in and surrounding tumor cells (*Figure 4E*), suggesting potential roles for *irf7* in inhibiting p53EPS initiation in tumor cells and/or within the TME.

Given broad Irf gene expression in p53EPS tumors (*Figure 2E*, *Figure 2—figure supplement 2*, *Figure 4—figure supplement 1*, *Supplementary file 5*), as well as reports of *IRF7* gene expression in human glioblastoma tumor cells, association with worsened patient outcome, and potential roles in tumor stem cell biology (*Jin et al., 2012*; *Yang et al., 2022*), we decided to investigate tumor cell-specific roles for Irf genes in p53EPS initiation. To specifically knock down *irf7* in p53EPS tumor cells, we generated a transgenic Tg(*her4.1*:Cas9-2A-EGFP);*tp53⁻ᐟ⁻* zebrafish strain and co-injected embryos with linearized EPS + gRNAs targeting *irf7* (*Figure 4F*). Despite a gene targeting efficiency of ~28% in pooled tumor cells from five tumor-burdened animals, we did not observe any significant changes in p53EPS tumor formation in Cas9-2A-EGFP-positive zebrafish compared to Cas9-2A-EGFP-negative control-injected siblings (*Figure 4F–H*). We observed similar effects following co-injection of gRNAs targeting *irf8* (*Figure 4G*), suggesting that increased p53EPS

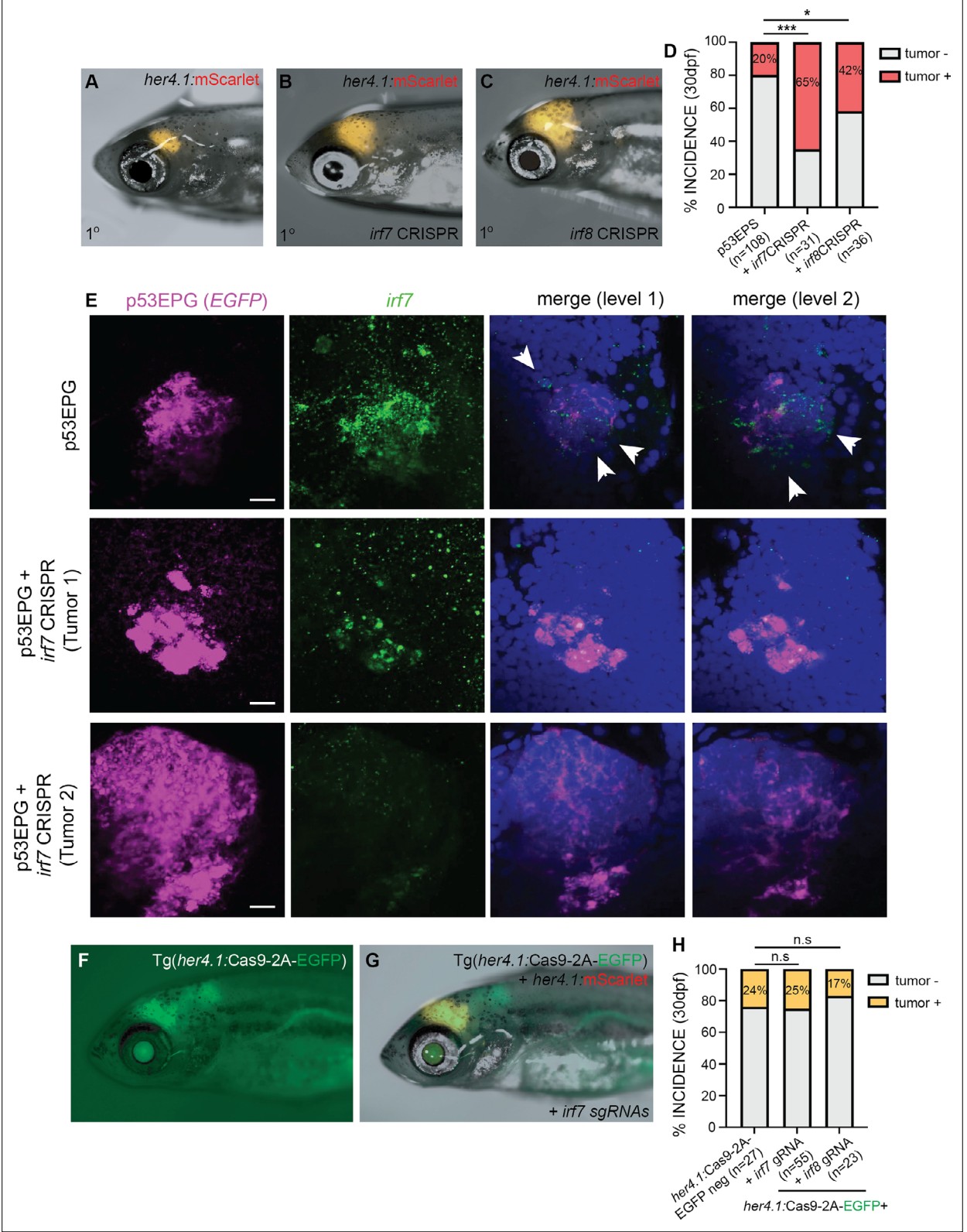

**Figure 4.** Inflammation-associated *irf7* and *irf8* inhibit p53EPS formation in vivo. (**A–C**) Primary (1°) control (**A**), *irf7* CRISPR/Cas9 (**B**), and *irf8* CRISPR/Cas9 (**C**) injected p53EPS at 30 days post fertilization (dpf). (**D**) p53EPS incidence at 30 dpf in control (n=3 independent experiments, 108 zebrafish), *irf7* CRISPR/Cas9 (***p<0.0001, Fisher's exact test, n=2 independent experiments, 31 zebrafish), and *irf8* CRISPR/Cas9 (*p=0.0155, Fisher's exact test, n=2 independent experiments, 36 total injected zebrafish). (**E**) Representative fluorescent in situ hybridization (FISH) images of whole mount control

*Figure 4 continued on next page*

*Figure 4 continued*

p53EPG (**E**) **E**GFR^vIII + PI3KCA^H1047 + **E**GFP and p53EPG + *irf7* CRISPR/Cas9-injected zebrafish at 8 dpf. p53EPG (EGFP, magenta) and *irf7* (green) images represent Z-stack projections through tumor lesions (11 optical sections each). Merged images represent single optical sections at two spatially separated levels within control and *irf7* knock-down tumors. DAPI staining (blue) is used to label nuclei. White arrowheads highlight *irf7* expression specific to the tumor microenvironment (TME). Scale bars represent 10 μm. (**F**) Tg(*her4.1*:Cas9-2A-EGFP) expression at 30 dpf. (**G**) mScarlet+ p53EPS at 30 dpf in Tg(*her4.1*:Cas9-2A-EGFP) injected with *irf7* guide RNAs (gRNAs) at the one-cell stage. (**H**) p53EPS incidence at 30 dpf in Tg(*her4.1*:Cas9-2A-EGFP)-negative gRNA-injected control siblings, and Tg(*her4.1*:Cas9-2A-EGFP) zebrafish injected at the one-cell stage with *irf7* or *irf8* gRNAs. n.s. not significant, Fisher's exact test.

The online version of this article includes the following figure supplement(s) for figure 4:

**Figure supplement 1.** Quantitative real-time PCR analysis of *irf7* CRISPR/Cas9-injected (**A**) and *irf8* CRISPR/Cas9-injected p53EPS (**B**).

**Figure supplement 2.** Neutral-red staining of phagocytic cell lineages in control and *irf8* CRISPR-injected zebrafish larvae.

tumor initiation following *Irf* gene knock-down is a consequence of *irf7* and *irf8* loss-of-function in the TME.

## Phagocyte activity suppresses p53EPS engraftment following transplant into syngeneic host zebrafish

Given conserved roles for Irf8 across species in the specification and function of different phagocyte populations (*Shiau et al., 2015*), our functional data suggested an anti-tumor role for *irf*/phagocyte activity in the TME during p53EPS tumor initiation. To assess the role more specifically for phagocyte populations within the TME, we decided to transplant p53EPS tumor cells into the hindbrain ventricle of syngeneic host embryos at 2 dpf. Importantly, serial transplantation of tumor cells is commonly used as an experimental surrogate to assess relapse potential (*Blackburn et al., 2011*; *Hayes et al., 2018*; *Ignatius et al., 2017*; *Smith et al., 2010*), allowing us to use p53EPS engraftment and re-growth to study the tumor propagating properties of p53EPS within different TMEs. To test the role for phagocytes in engraftment, we transplanted dissociated bulk tumor cells into syngeneic embryos along with Clodronate (Clodronate liposomes, Clodrosomes), a chemical used to eliminate phagocytes in vivo, and microglia following intracerebral injection (*Andreou et al., 2017*; *Hanlon et al., 2019*; *Yan et al., 2019*; *Yang et al., 2021*). At 18 dpt (20 dpf), we observed 50% p53EPS tumor cell engraftment in host animals co-injected with Clodronate liposomes, compared to 23% injected with vehicle control liposomes (p=0.0048, Fisher's exact test, *Figure 5A–C*). Interestingly, while p53EPS engrafted brains maintained inflammatory gene expression patterns like primary p53EPS tumor-burdened brains (*Figure 5—figure supplement 1*), co-transplantation with Clodronate liposomes inhibited inflammatory gene expression in bulk tissue (*Figure 5—figure supplement 1*, normalized to mScarlet expression to control for differences in tumor size), suggesting that reduced phagocyte-driven inflammation supports p53EPS tumor cell engraftment and growth. We also performed p53EPS bulk tumor cell transplantations into the hindbrain ventricles of *irf8* CRISPR/Cas9-injected syngeneic host zebrafish embryos at 2 dpf and observed 46% p53EPS tumor cell engraftment at 20 dpf, compared to 19% engraftment in *irf8* wild-type host zebrafish (p=0.0002, Fisher's exact test, *Figure 5D*), further supporting an inhibitory role for phagocytes within the TME. Altogether, our modeling data using patient-relevant oncogene combination in *tp53* loss-of-function background demonstrates an important role for inflammation in glioblastoma initiation and relapse including inhibitory roles for Irf-dependent signaling pathways, which in part may be attributed to anti-tumoral phagocyte activity within the TME.

## Discussion

Genetically engineered mouse models are widely used for the study of tumorigenesis in a physiological context; however, most cannot fully recapitulate the genetic heterogeneity found in glioblastoma in a timely and cost-effective manner and are therefore limited in the context of preclinical drug testing. Patient-derived xenografts (PDX) resemble patient tumors more closely as they retain mutational heterogeneity; however, PDX models cannot be used to address mechanisms of tumor onset in an intact and endogenous TME given that they are derived from pre-evolved tumor tissue and are either studied ex vivo or engrafted into immune-deficient animal hosts. Human stem cell-derived organoids represent a great advance in the field for the study of tumor development and

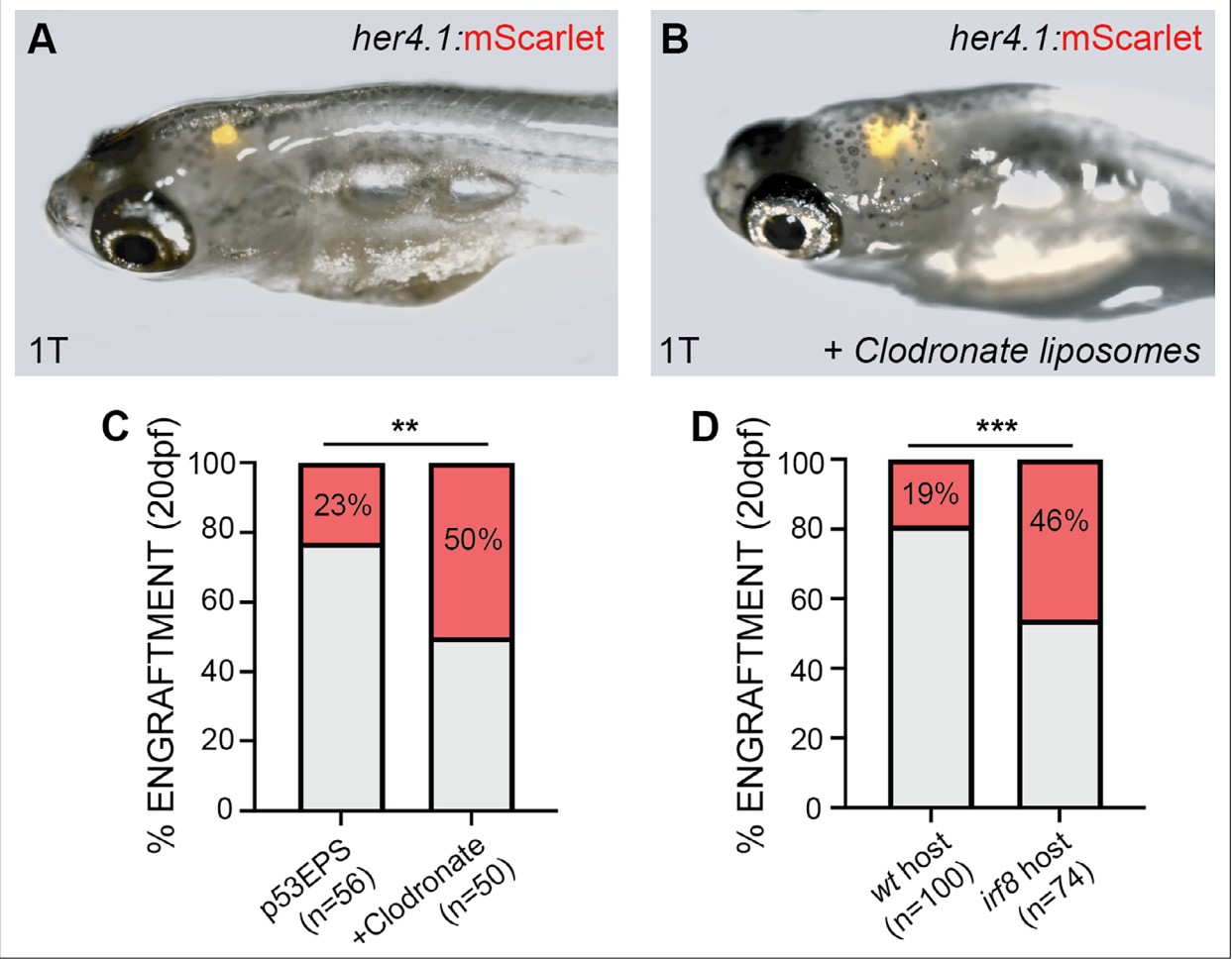

**Figure 5.** Inflammation-associated phagocytes inhibit p53EPS tumor engraftment. (**A, B**) CG1 syngeneic host zebrafish at 20 days post fertilization (dpf) engrafted with p53EPS tumor cells transplanted with vehicle control (**A**) or Clodronate liposomes (**B**) at 2 dpf. (**C–E**) Quantification of p53EPS control engrafted and p53EPS tumors engrafted into CG1 host embryos with (**C**) Clodronate liposomes (p=0.0048, Fisher's exact test, n=2 independent experiments, total 56 transplanted vehicle control and 50 transplanted Clodrosome-injected hosts), (**D**) engrafted into *irf8* CRISPR/Cas9-injected into CG1 syngeneic host embryos (p=0.0002, Fisher's exact test, n=2 independent experiments, total 100 transplanted control and 74 transplanted *irf8* CRISPR/Cas9-injected hosts).

The online version of this article includes the following figure supplement(s) for figure 5:

**Figure supplement 1.** Quantitative real-time PCR analysis of control, primary tumor, and engrafted whole brains.

for screening new therapeutic strategies, but the lack of an intact TME excludes both known and unknown intercellular interactions, which are increasingly appreciated to influence tumor cell evolution and drug responses in patients. Therefore, to address current limitations with respect to modeling human glioblastoma in vivo, we used zebrafish and neural stem/progenitor-specific expression of common oncogenic variants in a *tp53*-deficient background. We generated a robust tumor model, with a gene expression signature consistent with human glioblastoma of the mesenchymal subtype. We also harnessed the unique advantages of zebrafish modeling including in vivo live imaging and rapid genetic manipulations to assess characteristic inflammatory gene expression signatures within the TME, as well as roles for certain inflammatory cell types at tumor initiation and relapse following transplantation into syngeneic host animals.

Several animal models including zebrafish models of human brain cancer have been generated using transgenic over-expression of human oncogenes or xenograft transplantation of patient-derived tumor cells into immune-compromised hosts (*Almstedt et al., 2022*; *Ju et al., 2015*; *Mayrhofer et al., 2017*; *Modzelewska et al., 2016*), and these models have proven useful for defining important molecular pathways in glioma transformation, tumor growth, and cell migration. However, most of

these available zebrafish models are induced using single oncogenic drivers driven by a variety of neuronal and/or glial-specific promoters at different time points throughout development, with the use of single oncogenes in a functional *tp53* signaling background potentially contributing to their inability to model highly aggressive and relapse-like phenotypes in vivo. Xenograft transplantations of human glioblastoma cells into zebrafish larvae omit a fully functional immune system, which our work suggests plays an important role on tumor initiation and growth both in the primary context and following p53EPS tumor cell transplantation. Therefore, our zebrafish model of aggressive and transplantable human glioblastoma represents an important opportunity to study aspects related to tumor cell progression and relapse in an intact TME.

One important consideration with model generation is use of genetic drivers that are reflective of variants found in patient tumors. EGFR[viii] is a common genetic variant in glioblastoma that we show contributes to aggressive brain tumor formation in our zebrafish model. Interestingly, we found that our p53EPS model significantly enriches for gene expression signatures associated with mesenchymal glioblastoma, which may be unexpected given our use of EGFR[viii] as a driver (*Verhaak and Valk, 2010*). However, from our p53EPS transcriptional signatures, we also detected expression of several markers associated with EGFR[viii] and the classical subtype including neural progenitor and neural stem cell genes (*Figure 2—figure supplements 1 and 2*; *Supplementary file 1*), potentially suggesting underlying molecular heterogeneity. In addition, histological characterization of p53EPS tumors reveal several embryonal-type features, suggesting pathological differences with respect to over-expressing human oncogenes during zebrafish development, which likely affect certain morphological features and/or gene expression patterns compared to those seen in adult glioblastoma patients. A significant amount of heterogeneity is seen in patients with respect to gene expression and histopathology, and our characterizations are consistent with certain zebrafish and rodent models using MAPK/AKT pathway activation within the developing nervous system (*Huse and Holland, 2009*; *Jin et al., 2021*; *Mayrhofer et al., 2017*; *Wei et al., 2006*). However, in the future it will be interesting to use our model and assess temporal and driver-specific contributions to specific pathohistological features and molecular subtypes at early and late stages of tumor progression, as well as inter-tumoral heterogeneity and subtype transitions to better understand how phenotypic plasticity contributes to glioblastoma growth and relapse in our model, and in patients.

Zebrafish offer an in vivo platform for live imaging tumor cell dynamics at high resolution, and we demonstrated a powerful model to study intercellular interactions during glioblastoma initiation within an endogenous TME, an approach that is not currently feasible in rodent models. The contributions of macrophages/microglia and other phagocytes to an immunosuppressive TME in glioblastoma and their prevalence within the tumor bulk is well known and has made macrophages an attractive therapeutic target for patients. For example, CSF/CSF-1R interactions have been shown to induce an immunosuppressive M2 phenotype in glioblastoma-associated microglia/macrophages, and blockade of CSF-1R improves survival in tumor-burdened mice (*Kennedy et al., 2013*; *Pyonteck et al., 2013*). However, our data and work from others suggests an equally important tumor-suppressive role for microglial/macrophages based on tumor stage and activation state. For example, serial transplantation of transformed mouse neural stem cells results in strong negative pressure from the immune system in syngeneic mice, leading to activation of tumor cell-intrinsic immune evasion mechanisms that support growth in vivo and in vitro (*Gangoso et al., 2021*). In addition, recent spatial analyses of human glioblastoma tumor samples identified regions of activated (myeloperoxidase) macrophages that are associated with improved clinical outcomes, suggesting anti-tumorigenic roles for certain macrophage populations in patients (*Karimi et al., 2023*). Our modeling further supports a strong anti-tumor role for pro-inflammatory phagocytes within the endogenous TME at initiation as well as following transplantation, which is considered an experimental surrogate for relapse potential. Our data also suggests that the pro-tumorigenic immune niche seen in later-stage gliomas is not already established at initiation stages. Given that the transition to mesenchymal-like glioblastoma is closely linked to immune cell infiltration and an immune-suppressive TME (*Gangoso et al., 2021*; *Klemm et al., 2020*), it will be important to characterize these transitions to better understand anti-tumor mechanisms and their evolution during tumor progression, with the potential for therapeutically harnessing endogenous immune cell types and/or signaling pathways in the future. Altogether, these data suggest an anti-tumor growth but potentially relapse promoting role for therapeutic anti-macrophage approaches, highlighting the importance of future in vivo investigations into how

inhibition of transitions and/or switching immunosuppressive microglia/macrophages to immune activating states may inhibit glioblastoma growth or prevent relapse for patients. Given the role for a complexity of cell types in inflammation, it will be interesting to further assess both the innate and adaptive immune systems. For example, based on common markers we observed less evidence for neutrophils within p53EPS, consistent with the relatively low presence of neutrophils seen in human glioblastoma (*Friedmann-Morvinski and Hambardzumyan, 2023*). However, how these and other cell types change depending on specific genetic drivers and/or tumor stage should be highly accessible and translatable using our zebrafish model system given the broad availability of validated reporter transgenes, as well as the high level of genetic and physiological conservation with human.

Finally, it is well known that the type of glioma affects immune responses, implying that tumor-intrinsic factors shape the composition of the TME (*Friebel et al., 2020*; *Gangoso et al., 2021*; *Klemm et al., 2020*). Therefore, phenotypic differences with respect to glioblastoma-TME interactions in patients and model systems are likely influenced by the genetic background and oncogenic driver events. Human AKT1 over-expression in zebrafish neural cells was previously shown to drive pre-neoplastic lesions in zebrafish larvae and was found to recruit macrophage and microglia populations through the Sdf1b-Cxcr4b signaling pathway (*Chia et al., 2018*). In this model, loss of macrophages resulted in decreased oncogenic proliferation, suggesting tumor-promoting functions for macrophages at initiation stages. In contrast to AKT1-expressing pre-neoplastic neurons, we visualized that neutral red-positive phagocytes and *mpeg1.1*:GFP+ macrophages/microglia very closely associated with and/or engulfed mScarlet+ p53EPS tumor cellular puncta, suggesting an effort to clear cells and/or cellular debris in p53EPS. This anti-tumor role is further supported by our functional analyses showing increased p53EPS tumor initiation in the context of Irf8 gene knock-out and decreased phagocytic cell development in vivo in zebrafish larvae. Given that effectively leveraging targeted therapies for glioblastoma will require a deep and individualized understanding of patient-specific tumor cell biology, it will be important to harness the flexibility of our mosaic oncogene over-expression approach and understand different tumor plus environmental factors in the context of different genetic drivers and/or modifiers identified in patients.

# Methods

All materials created in this study are available upon request to madeline.hayes@sickkids.ca.

## Key resources table

| Reagent type (species) or resource | Designation | Source or reference | Identifiers | Additional information |
|---|---|---|---|---|
| Strain, strain background (*zebrafish*) | CG1 | *Mizgireuv and Revskoy, 2006* PMID:16540662 | | |
| Strain, strain background (*zebrafish*) | CG1tp53-/- | *Ignatius et al., 2018* PMID:30192230 | | |
| Strain, strain background (*zebrafish*) | Tg(*mpeg1.1*:EGFP) | *Ellett et al., 2011* PMID:21084797 | | |
| Strain, strain background (*zebrafish*) | Tg(*tnfa*: EGFP) | *Nguyen-Chi et al., 2015* PMID:26154973 | | |
| Transfected construct (plasmid, injected) | *her4.1*:PI3KCA[H1047R] | This paper | | Gateway cloning using 5' zebrafish her4.1 promoter construct (*Yeo et al., 2007*, PMID:17134690), middle entry PI3KCA(H1047R) ORF, and 3' polyA into a Tol2-compatible Destination vector (Tol2kit, *Kwan et al., 2007*, PMID:17937395). Plasmid was linearized for injection using Xho1 restriction enzyme. |
| Transfected construct (plasmid, injected) | *her4.1:* mScarlet | This paper | | Gateway cloning using 5' zebrafish her4.1 promoter construct (*Yeo et al., 2007*, PMID:17134690), middle entry mScarlet ORF, and 3' polyA into a Tol2-compatible Destination vector (Tol2kit, *Kwan et al., 2007*, PMID:17937395). Plasmid was linearized for injection using Xho1 restriction enzyme. |
| Transfected construct (plasmid, injected) | *her4.1:* EGFR[vIII] | This paper | | Gateway cloning using 5' zebrafish her4.1 promoter construct (*Yeo et al., 2007*, PMID:17134690), middle entry EGFR(vIII) ORF, and 3' polyA into a Tol2-compatible Destination vector (Tol2kit, *Kwan et al., 2007*, PMID:17937395). Plasmid was linearized for injection using Xho1 restriction enzyme. |
| Transfected construct (plasmid, injected) | *her4.1:* KRAS[G12D] | This paper | | Gateway cloning using 5' zebrafish her4.1 promoter construct (*Yeo et al., 2007*, PMID:17937395), middle entry KRAS(G12D) ORF, and 3' polyA into a Tol2-compatible Destination vector (Tol2kit, *Kwan et al., 2007*, PMID:17937395). Plasmid was linearized for injection using Xho1 restriction enzyme. |

*Continued on next page*

*Continued*

| Reagent type (species) or resource | Designation | Source or reference | Identifiers | Additional information |
|---|---|---|---|---|
| Transfected construct (plasmid, injected) | *her4.1:* EGFP | This paper | | Gateway cloning using 5' zebrafish her4.1 promoter construct (*Yeo et al., 2007*, PMID:17134690), middle entry EGFP ORF, and 3' polyA into a Tol2-compatible Destination vector (Tol2kit, *Kwan et al., 2007*, PMID:17937395). Plasmid was linearized for injection using Xho1 restriction enzyme. |
| Transfected construct (plasmid, injected) | *her4.1:* Cas9-2A-EGFP | This paper | | Gateway cloning using 5' zebrafish her4.1 promoter construct (*Yeo et al., 2007*, PMID:17134690), middle entry Cas9 ORF, and 3' 2A-EGFP into a Tol2-compatible Destination vector (Tol2kit, *Kwan et al., 2007*, PMID:17937395). Plasmid was injected with Tol2 transposase mRNA to establish stable transgenic lines. |
| Transfected construct (plasmid, injected) | *gfap:* EGFP | This paper | | Gateway cloning using 5' zebrafish gfap promoter construct (*Don et al., 2017*, PMID:27631880), middle entry EGFP ORF, and 3' polyA into a Tol2-compatible Destination vector (Tol2kit, *Kwan et al., 2007*, PMID:17937395). Plasmid was linearized for injection using Cla1 restriction enzyme. |
| Transfected construct (plasmid, injected) | *gfap:* mScarlet | This paper | | Gateway cloning using 5' zebrafish gfap promoter construct (*Don et al., 2017*, PMID:27631880), middle entry mScarlet ORF, and 3' polyA into a Tol2-compatible Destination vector (Tol2kit, *Kwan et al., 2007*, PMID:17937395). Plasmid was linearized for injection using Cla1 restriction enzyme. |
| Transfected Construct (plasmid, injected) | *mpeg1.1:* EGFP | *Ellett et al., 2011* PMID:21084797 | | Plasmid was linearized for injection using Xho1 restriction enzyme. |
| Transfected construct (plasmid, injected) | *rag2:* EGFP | *Langenau et al., 2007* PMID:17510286 | | Plasmid was linearized for injection using Xho1 restriction enzyme. |
| Antibody | Anti-PCNA (rabbit monoclonal) | Cell Signaling | D3H8P | Antibody was used for IHC at 1/200 dilution |
| Antibody | Anti-phospho-p44/42 MAPK (Erk1/2) (Thr202/Tyr204) (rabbit polyclonal) | Cell Signaling | 9101 | Antibody was used for IHC at 1/200 dilution |
| Antibody | Anti-phospho-AKT (Ser473) (rabbit polyclonal) | Cell Signaling | 9271 | Antibody was used for IHC at 1/300 dilution |
| Sequence-based reagent | EGFP HCR probe | Clontech | Molecular Instruments | |
| Sequence-based reagent | Irf7 HCR probe | Sigma-Aldrich | Molecular Instruments | |
| Peptide, recombinant protein | Cas9 protein with NLS | PNA-Bio | CP01-200 | |
| Commercial assay of kit | In vitro sgRNA synthesis kit | New England Biolabs | E3322V | Individual sgRNA target sites are indicated in *Supplementary file 7* |
| Chemical compound, drug | Clodronate lipsomes (Clodrosomes) | Encapsula Nano Sciences | CLD-8901 | |

## Zebrafish husbandry and care

Animals were raised in accordance with Canadian Council on Animal Care (CCAC) guidelines and all experiments were approved under an Animal Use Protocol established with the Animal Care Committee at the Hospital for Sick Children Research Institute (AUP #1000051391 and #1000064586). Previously described zebrafish strains including syngeneic CG1 (*Mizgireuv and Revskoy, 2006*), CG1tp53[del] (*Ignatius et al., 2018*), Tg(*mpeg1.1*:EGFP) (*Ellett et al., 2011*), and Tg(*tnfa*:EGFP) (*Nguyen-Chi et al., 2015*) were used as indicated in the manuscript.

## Preparation and injection of linearized DNA for tumorigenesis

*her4.1*:PI3KCA[H1047R], *her4.1*:mScarlet, *her4.1*:EGFR[vIII], *her4.1*:KRAS[G12D], *her4.1*:EGFP *her4.1*:Cas9-2A-EGFP, *gfap*:EGFP, and *gfap*:mScarlet transgene expression constructs were cloned using previously described *her4.1* sequence (*Yeo et al., 2007*), *gfap* entry plasmid (*Don et al., 2017*), standard Tol2 Gateway plasmids and protocols (*Kwan et al., 2007*). *rag2*:EGFP and *mpeg1.1*:EGFP expression plasmids were previously described (*Ellett et al., 2011*; *Langenau et al., 2007*). Circular *her4.1* expression plasmids were linearized using XhoI restriction enzyme (New England Biolabs, R0146S) while *gfap*:EGFP was linearized using ClaI (New England Biolabs, R0197S), according to the manufacturer's

protocol. Restriction enzymes were heat inactivated and linearized vector purified utilizing an EZ-10 Spin Column PCR Products Purification Kit (Bio Basic, BS364). Injection mixtures such as EPS (*her4.1*:**E**GFR$^{vIII}$, *her4.1*:**P**I3KCA$^{H1047R}$, *her4.1*:m**S**carlet) or KPG (*her4.1*:**K**RAS, *her4.1*:**P**I3KCA$^{H1047R}$, *her4.1*:**G**FP) were prepared at 2:1:1 molar ratios in 50% TE Buffer (Invitrogen, 12090-015) with KCl (final concentration 0.1 M) and phenol red (final concentration 5%, Sigma-Aldrich, P0290). CG1 or CG1tp53$^{del}$ embryos were microinjected with 0.5–1 nL of injection mixture at the one-cell stage and monitored for tumor development, starting at 10–15 dpf.

## gRNA synthesis and CRISPR-Cas9-mediated gene targeting

CRISPR/Cas9 sequence targets and gRNA oligos were designed using CHOPCHOP (*Labun et al., 2019*). gRNAs were in vitro synthesized using the EnGen sgRNA Synthesis Kit (New England Biolabs, E3322V) according to the manufacturer's recommendation. Cas9 protein with NLS (PNA Bio, CP01-200) was resuspended in 20% glycerol/water to a concentration of 1 mg/mL. Cas9/gRNA microinjection mixture was prepared at a final concentration of 0.3 mg/mL Cas9+30–50 ng/µL of each gRNA. 0.5–1 nL of injection mixture injected into zebrafish embryos of the indicated genotype, at the one-cell stage. CRISPR/Cas9 targeting efficiency was measured following PCR-based locus amplification, Sanger sequencing, and Synthego ICE Analysis (ICE Analysis, Synthego, https://ice.synthego.com/; *Synthego, 2019*). All gRNA sequences and PCR oligos are indicated in *Supplementary file 7*.

## Brain dissection and dissociation

Animals were euthanized with a lethal 300 mg/L dose of Tricaine (Sigma-Aldrich, E10521) ~10–20 min before dissection. Fish were decapitated posterior of the gills and the head transferred to sterile-filtered PBS (Wisent Inc, 311-010-CL). Using fine-tipped forceps, brains were carefully extracted from the skull and transferred to a microcentrifuge tube and kept in PBS at 28°C until further processing. 5–10 brains were transferred to 1 mL of 28°C pre-warmed Accutase (STEMCELL Technologies, 07920), followed by incubation at 28°C with gentle rocking for a total of ~50 min. Every 10 min, brains were mechanically dissociated with gentle pipetting 15–30 times using a 1 mL filter tip. Dissociated tissues were passed through a 40 µm Cell Strainer (Corning, 352340) into a 50 mL conical tube to achieve single-cell suspensions. Strained cells were pelleted at approximately 1000 × *g* for 5 min at room temperature and resuspended in PBS.

## Bulk tumor cell transplantation

Cell suspensions were maintained at 28°C during the transplantation procedure. 2 dpf CG1 strain syngeneic zebrafish larvae were injected with 1–2 nL of cell suspension into the hindbrain ventricle, as previously described (*Casey et al., 2017*). Cells were injected alone or in combination with 1% total volume Clodrosomes (Encapsula Nano Sciences, CLD-8901) or vehicle control liposomes.

## Immunohistochemistry and FISH

Zebrafish were fixed overnight in 4% PFA and stored in methanol at –20°C before paraffin embedding and sectioning at the Centre for Phenogenomics Pathology Core Facility. Animal sections were deparaffinized with 2×5 min washes of xylenes. Sections were rehydrated with sequential 2×10 min washes in 100%, 90%, 70% ethanol before sequential rinsing with ddH$_2$O, 3% H$_2$O$_2$, and ddH$_2$O. Slides were then boiled in 1× citrate buffer (Sigma-Aldrich, C999) within a standard microwave and rinsed in ddH$_2$O, after cooling for 30 min. Sections were blocked with TBST/5% Normal Goat Serum for 1 hr at room temperature before overnight incubation with primary antibody diluted in Diluent CST (Cell Signaling Technology, 8112) at 4°C. The following day slides were washed once with 3×5 min with TBS, TBST, and TBS. Samples were then incubated with secondary antibody (Cell Signaling Technology, 8114) for half an hour at room temperature, followed by another 3×5 min TBS/T wash. Standard DAB development (Thermo Scientific, 34002) was performed for 5–7 min, followed by water termination. Samples were Hematoxylin (VWR, 10143-146) stained with a 1:6 diluted solution for 3–5 s and rinsed 5× with tap water. Slides were dehydrated sequentially with 70%, 90%, 100% ethanol washes followed by 2×5 min xylene washes before being mounted with glass cover slips and Permount (Fisher Scientific, SP15-100). Primary antibodies raised against proliferating cell nuclear antigen (PCNA, Cell Signaling, D3H8P), p-ERK (Cell Signaling, #9101), and p-AKT (Cell Signaling, #9271) were used.

Whole mount FISH was performed on fixed larvae using *EGFP* and *irf7*-specific antisense probes (Molecular Instruments) and HCR protocol, according to the manufacturer's recommendations.

## Bulk RNAseq library preparation, quantification, and differential gene expression analysis

Three p53KPG, p53EPS, and control-injected brains (from non-tumor forming injected siblings) were harvested at 20–30 dpf and immediately placed into 1 mL of TRIzol Reagent (Invitrogen, 15596026). mScarlet+ tumor cells were also sorted from bulk (non-fluorescent) tissue from pooled brains at the SickKids-UHN Flow Cytometry Core Facility on a Sony MA900 VBYR cell sorter, before pelleting and lysis in TRIzol Reagent. Total RNA was purified for all samples using a Monarch RNA Cleanup Kit (New England Biolabs, T2040L), according to the manufacturer's recommendations. Sequence ready polyA-enriched libraries were prepared using the NEB Ultra II Directional mRNA prep kit for Illumina (NEB, E7760), according to the manufacturer's recommendations. Single-end 150 bp sequencing at a targeted depth of ~30–60 million reads/sample was performed using an Illumina NovaSeq S1 flowcell, at the Centre for Applied Genomics (TCAG). Raw .fastq data was processed using Salmon quantification of transcripts for each sample. A 'decoy-aware' index was built with the *Danio rerio* transcriptome and genome using the GRCz11 assembly with a k-mers length of 23. Samples were then quantified with the following arguments: -r, `--seqBias`, `--mp` –3, `--validateMappings`, `--rangeFactor-izationBins` 4. Sequencing data is available from GEO under accession code GSE246295.

## Human ortholog conversion and GSEA

Normalized counts output from DESeq2 for the above processed data were utilized for GSEA without any further trimming or processing, as recommended by the GSEA user guide (*Mootha et al., 2003*; *Subramanian et al., 2005*). Zebrafish transcripts were assigned known or high-confidence human orthologs using Ensembl BioMart, as previously described (*Demirci et al., 2022*). Bulk sequencing data from *her4.1*:KRAS^G12D or *her4.1*:EGFR^vIII-driven tumors, or sorted cells derived from such, were compared with non-tumour control brains under default conditions (1000 permutations, gene set). Expression signatures were compared against published glioblastoma (*McLendon et al., 2008*; *Wang et al., 2017*), medulloblastoma (*Cavalli et al., 2017*), Hallmark, and/or C7: immunologic signature gene sets (*Villanueva et al., 2011*), as indicated in the manuscript.

## Neutral red staining and quantification

Zebrafish were treated with neutral red, as previously described (*Shiau et al., 2013*). Animals were then anesthetized in Tricaine and oriented on their sides in 3% methylcellulose before an image stack of approximately 120–150 µm depth was taken beginning from the surface of the otic vesicle into the fish. Neutral red stained foci were counted using Zen Lite Software (version 3.3) in an area bounded by the posterior edge of the eye and the posterior edge of the otic vesicle for each fish.

## Image acquisition

Fluorescence and bright-field image acquisition of whole animals and dissected tumor-burdened brains was performed using a Zeiss Axiozoom V16 macroscope. High-resolution confocal microscopy was performed using a Nikon 1AR confocal microscope, with image processing and Z-stack compression performed using ImageJ software. IHC slides were imaged on a Pannoramic Flash II.

## RT-qPCR and analysis

Total RNA was extracted from tumor-burdened brains using TRIzol reagent, as described above, and reverse-transcribed using the High-Capacity DNA Reverse Transcriptase Kit (Thermo Fisher, 4368814), as per the manufacturer's recommendations. RT-qPCR was performed using the SYBR Green I Master kit (Roche, 04887352001) following the manufacturer's recommended protocol and a Bio-Rad CFX96 qPCR Real-Time PCR Module with C1000 Touch Thermal Cycler Unit. All primers used to amplify genes of interest are indicated in *Supplementary file 7*. At least five brain samples were pooled for each sample, with gene expression normalized to *18s*. Where indicated, gene expression was normalized to *mScarlet* expression to control for differences in overall tumor size.

## Acknowledgements

We thank Dr. David Langenau for CG1*tp53*del zebrafish. We also thank the Centre for Phenogenomics (TCP) Histopathology core for tissue processing and staining, the Centre for Applied Genomics (TCAG) for sequencing, the SickKids Imaging Facility, and the Zebrafish Genetics and Disease Models Facility management and support team for zebrafish husbandry. We thank Dr. Matt Wood at Oregon Health & Science University for pathology on zebrafish brain tumors.

## Additional information

### Funding

| Funder | Grant reference number | Author |
|---|---|---|
| Ontario Institute for Cancer Research | IA-026 | Alex Weiss<br>Bret J Pearson |
| Canadian Institutes of Health Research | PJT-159611 | Cassandra D'Amata |
| Oregon Health and Science University | Start-up Funds | Bret J Pearson |
| Hospital for Sick Children | Start-up Funds | Madeline N Hayes |
| Canadian Institutes of Health Research | Fellowship | Madeline N Hayes |
| National Institutes of Health | 1R01GM155244 | Bret J Pearson |

The funders had no role in study design, data collection and interpretation, or the decision to submit the work for publication.

### Author contributions

Alex Weiss, Conceptualization, Data curation, Formal analysis, Investigation, Methodology, Writing – review and editing; Cassandra D'Amata, Formal analysis, Investigation, Methodology, Writing – review and editing; Bret J Pearson, Conceptualization, Resources, Supervision, Funding acquisition, Methodology, Writing – original draft, Project administration, Writing – review and editing; Madeline N Hayes, Conceptualization, Data curation, Formal analysis, Supervision, Funding acquisition, Investigation, Methodology, Writing – original draft, Writing – review and editing

### Author ORCIDs

Bret J Pearson ORCID http://orcid.org/0000-0002-3473-901X
Madeline N Hayes ORCID https://orcid.org/0000-0002-0089-8995

### Ethics

This study was performed in strict accordance with Canadian Council on Animal care (CCAC) guidelines and all experiments were approved under an Animal Use Protocol established with the Animal Care Committee at the Hospital for Sick Children Research Institute (AUP #1000051391 and #1000064586).

Reviewer #1 (Public review): https://doi.org/10.7554/eLife.93077.3.sa1
Reviewer #2 (Public review): https://doi.org/10.7554/eLife.93077.3.sa2
Author response https://doi.org/10.7554/eLife.93077.3.sa3

## Additional files

### Supplementary files

• Supplementary file 1. Differential gene expression (from DESeq2) of p53EPS tumor-burdened whole brain samples compared to tumor-negative samples (n=3 independent samples per cohort). Genes with 1>log2foldChange>–1, adjusted p-value<0.05 are shown.

• Supplementary file 2. Differential gene expression (from DESeq2) of p53EPS tumor-burdened

whole brain samples, compared to tumor-negative samples (n=3 independent samples per cohort) with corresponding human homologues. Genes with 1>log2foldChange>–1, adjusted p-value<0.05 are shown.

• Supplementary file 3. Results of gene set enrichment analysis (GSEA) for published glioblastoma subtype gene sets (*McLendon et al., 2008*; *Wang et al., 2017*).

• Supplementary file 4. Results of gene set enrichment analysis (GSEA) for Hallmark gene sets (*Villanueva et al., 2011*).

• Supplementary file 5. Differential gene expression (from DESeq2) of mScarlet-sorted p53EPS tumor cells, compared to tumor-negative whole brain samples (n=1 pooled sorted cell sample versus n=3 independent tumor-negative controls).

• Supplementary file 6. Results of gene set enrichment analysis (GSEA) for C7: immunologic signature gene sets and GSE5099 gene sets from the Molecular Signatures Database (*Villanueva et al., 2011*).

• Supplementary file 7. List of oligo sequences used for quantitative RT-PCR, sgRNA synthesis, genotyping, and gene target validation.

• MDAR checklist

## Data availability

Sequencing data have been deposited in GEO under accession code GSE246295. Where indicated, source data is provided as supplementary files.

The following dataset was generated:

| Author(s) | Year | Dataset title | Dataset URL | Database and Identifier |
|---|---|---|---|---|
| Hayes M, Pearson B, Weiss A, D'Amata C | 2024 | A syngeneic spontaneous zebrafish model of tp53-deficient, EGFRviii, and PI3KCAH1047R-driven glioblastoma reveals inhibitory roles for inflammation during tumor initiation and relapse in vivo | https://www.ncbi.nlm.nih.gov/geo/query/acc.cgi?acc=GSE246295 | NCBI Gene Expression Omnibus, GSE246295 |

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
