## [Editor Report · eLife assessment]

This study presents a **valuable** syngeneic zebrafish model for studying glioblastoma and will be of interest to neuro-oncologists and cancer biologists. Using a feasible in vivo model to study the tumour microenvironment, cell/cell interaction, and immunity, the data are **compelling**, and opens up new lines of inquiries for future investigation on the impact of efferocytosis on tumor progression and cell of origin in this model as well as assessments of drug resistance mechanisms, using inhibitors to MAPK , Akt and/or mTOR pathway.

---

## [Referee Report · Reviewer #1 (Public review)]

Summary:

The authors have developed a zebrafish model of glioblastoma and characterized this, with a particular focus on the role of recruited myeloid cells in the tumours. Microglia/macrophages in the tumours are proposed to have an inflammatory phenotype and are engaged in phagocytosis. Knockout of Irf7 and Irf8 genes enhanced tumour initiation. Depleting mature myeloid cell types with chlodronate also enhanced tumour initiation. It is proposed that in early stage tumours, microglia/macrophages have tumour suppressive activity.

Strengths:

The authors have generated a novel glioblastoma model in zebrafish. Two key strengths of the zebrafish model are that early stage tumours can be studied and in vivo visualization can be readily performed. The authors show video of microglia/macrophages adopting the ameboid phenotype in tumours (as is observed in human tumours) and engaging in phagocytosis. Video 1 was very impressive in my opinion and shows the model is a very useful tool to study microglia/macrophage:glioblastoma cell interactions. The irf7/irf8 knockdown and the chlodronate experiments are consistent with a role for mature myeloid cells in suppressing tumour initiation, suggesting that the model may also be very valuable in understanding immune surveillance in glioblastoma initiation.

Weaknesses:

EGFRvIII is mainly associated with the classical subtype, so the mesenchymal subtype might be unexpected here. This could be commented on. Some more histologic characterization of the tumours would be helpful. Are they invasive, do larger tumours show necrosis and microvascular proliferation? This would help with understanding the full potential of the new model. Current thinking in established human glioblastoma is that the M1/M2 designations for macrophages are not relevant, with microglia macrophage populations showing a mixture of pre- and anti-inflammatory features. Ideally there would be a much more detailed characterization of the intratumoral microglia/macrophage population here, as single markers can't be relied upon. Phagocytosis could have antitumour effects through removal of live cancer cells, or could be cancer promoting if apoptotic cancer cells are being rapidly cleared with concomitant activation of an immunosuppressive phenotype in the phagocytes (i.e. efferocytosis). It may be possible to distinguish between these two types of phagocytosis experimentally. Do the irf7/8 and chlodronate experiments distinguish between effects on microglia/macrophages and dendritic cells?

Update: The more detailed description of the tumour histology is very interesting and the authors have addressed my previous concerns nicely.

---

## [Referee Report · Reviewer #2 (Public review)]

Summary:

Glioblastoma is a common primary brain cancer, that is difficult to treat and has a low survival rate. The lack of genetically tractable and immunocompetent vertebrate animal model has prevented discovery of new therapeutic targets and limited efforts for screening of pharmaceutical agents for the treatment of the disease. Here Weiss et al., express oncogenic variants frequently observed in human glioblastoma within zebrafish lacking the tumor suppressor TP53 to generate a patient-relevant in vivo model. The authors demonstrate that loss of TP53 and overexpression of EGFR, PI3KCA, and mScarlet (p53EPS) in neural progenitors and radial glia leads to visible fluorescent brain lesions in live zebrafish. The authors performed RNA expression analysis that uncovered a molecular signature consistent with human mesenchymal glioblastoma and identified gene expression patterns associated with inflammation. Live imaging revealed high levels of immune cell infiltration and associations between microglia/macrophages and tumor cells. To define functional roles for regulators of inflammation on specific immune-related responses during tumorigenesis, transient CRISPR/Cas9 gene targeting was used to disrupt interferon regulator factor proteins and showed Inflammation-associated irf7 and irf8 are required to inhibit p53EPS tumor formation. Further, experiments to deplete the macrophages using clodronate liposomes suggest that macrophages contribute to the suppression of tumor engraftment following transplantation. The authors' conclusions are supported by the data and the experiments are thoroughly controlled throughout. Taken together, these results provide new insights into the regulation of glioblastoma initiation and growth by the surrounding microenvironment and provide a novel in vivo platform for the discovery of new molecular mechanisms and testing of therapeutics.

Strengths/Weaknesses:

The authors convincingly show that co-injection of activated human EGFRviii, PI3KCAH1047R, and mScarlet into TP53 null zebrafish promotes formation of fluorescent brain lesions and glioblastoma-like tumor formation. The authors include histological characterization of the tumors, as well as quantifications of p-ERK and p-AKT staining to highlight increased activation of the MAPK/AKT signaling pathways in their tumor model.

The authors use a transplantation assay to further test the tumorigenic potential of dissociated cells from glial-derived tumors in the context of specific manipulations of the tumour microenvironment.

The authors nicely show high levels of immune cell infiltration and associations between microglia/macrophages and tumor cells. Quantification of the emergence of macrophages over time in relation to tumor initiation and growth is provided and supports the observations of tumor suppressive activity of the phagocytes. The authors also attempt to delineate if other leukocyte populations are involved and observe tumor formation without significant infiltration of neutrophils.

The authors provide evidence for key genetic regulators of the local microenvironment, showing increased p53EPS tumor initiation following Ifr7 gene knock-down and loss of irf7 expression in the TME.

---

## [Author Response]

The following is the authors’ response to the original reviews.

**Reviewer #1:**
“EGFRvIII is mainly associated with the classical subtype, so the mesenchymal subtype might be unexpected here. This could be commented on.”

We acknowledge that EGFRvIII is most often associated with the classical subtype of glioblastoma and agree that mesenchymal subtype classification may be unexpected given the use of *her4.1*:EGFRvIII as a driver in our model. We would like to highlight the fact that our brain tumors do also express certain markers associated with the classical subtype including neural precursor and neural stem cell markers like *sox2, ascl1b,* and *gli2* (Supplementary Fig 4, 5; Supplementary Table 1-3). However, our transcriptomic data was not found to significantly enrich for classical subtype gene expression, compared to normal brains. This could be due to a significant contribution of normal brain tissue to our analyses (bulk tumor burdened brains were harvested for RNA sequencing), as well as the significant contribution of mesenchymal subtype signatures and/or inflammatory gene expression in our brain tumor-positive samples. Because signatures associated with inflammation consist of some of the most highly upregulated genes in our samples, this could potentially dilute out and/or lessen alterative subtype and/or signature gene expression. Importantly, it is now widely appreciated that patient tumors simultaneously consist of heterogenous tumor cells reflecting multiple molecular subtypes (Couturier et al., 2020; Darmanis et al., 2017; Neftel et al., 2019), providing glioblastoma with a high level of phenotypic plasticity. We also demonstrate that the contribution of additional drivers not always present with EGFRvIII in patient glioblastoma enhances primary brain tumors in vivo. This result is consistent with more aggressive glioblastomas seen in patients with EGFRvIII variants and TP53 loss-of-function mutations (Ruano et al., 2009). It will therefore be interesting in the future to consider how single or multiple driver mutations contribute to subtype-specific gene expression in our model, as well as histopathology, relative to patients. We have included some of these discussion points to our revised manuscript.

“Some more histologic characterization of the tumors would be helpful. Are they invasive, do larger tumors show necrosis and microvascular proliferation? This would help with understanding the full potential of the new model.”

We have updated our manuscript to include more histolopathological characterization and images (Supplementary Fig 2).

“Current thinking in established glioblastoma is that the M1/M2 designations for macrophages are not relevant, with microglia macrophage populations showing a mixture of pre- and anti-inflammatory features. Ideally, there would be a much more detailed characterization of the intratumoral microglia/macrophage population here, as single markers can’t be relied upon.”

We performed additional gene set enrichment analyses (GSEA) using our sequencing datasets and compared p53EPS gene expression to M1/M2 macrophage expression signatures and expression signatures from MCSF-stimulated macrophages at early and late (M2 polarized) time-points. From this analysis, we detected enrichment for markers of both pro- and antiinflammatory features, however, with stronger and significant enrichment for gene expression signatures associated with classical pro-inflammatory M1 macrophages. We have included these GSEA plots and gene set enrichment lists as supplementary materials (Supplementary Fig 6, Supplementary Table 6). We also performed GSEA against a broad curated set of immunologic gene sets (C7: immunologic signature gene sets, Molecular Signatures Database, (Liberzon et al., 2011)) and have included the list of signatures and enrichment scores as a supplementary table (Supplementary Table 6).

“Phagocytosis could have anti-tumor effects through removal of live cancer cells or could be cancer-promoting if apoptotic cells are being rapidly cleared with concomitant activation of an immunosuppressive phenotype in the phagocytes (ie. efferocytosis).”

We looked at efferocytosis-associated gene expression in our sequencing dataset (124 “efferocytosis” genes, GeneCards), and while we detected upregulation of certain genes associated with efferocytosis in p53EPS brains, we did not detect significant enrichment for the entire gene set. Furthermore, we did not detect up-regulation of key efferocytosis receptors including Axl and Tyro3 (Supplementary Table 1, 2), compared to normal brains. While efferocytosis may contribute to tumor growth and evolution, this GSEA combined with our functional data supporting an inhibitory role for phagocytes in p53EPS tumor initiation and engraftment following transplantation (Fig 4, Fig 5, Supplementary Fig 7), suggests that efferocytosis is not a major driver of tumor formation in our model. However, how efferocytosis affects tumor progression in our model and/or relapse following therapy will be an interesting feature to explore in the future using temporal manipulations of phagocytes and/or treatments with chemical inhibitors.

**Author response image 1. sa3fig1:** Gene Set Enrichment Analysis (GSEA) for efferocytosis-associated gene expression (124 “efferocytosis” genes in GeneCards) in tp53EPS tumor brains, compared to normal zebrafish brains. Normalized enrichment score (NES) and p-value are indicated.

“Do the irf7/8 and chlodronate experiments distinguish between effects on microglia/macrophages and dendritic cells?”

In addition to microglia/macrophages, the IRF8 transcription factor has been shown to control survival and function of dendritic cells (Sichien et al., 2016). Chlodronate treatments are also used to deplete both macrophages and dendritic cells in vivo. Therefore, we cannot distinguish the effects of these manipulations in our experiments and have updated our manuscript throughout to reflect this.

**Reviewer #2:**
“The authors state that oncogenic MAPK/AKT pathway activation drives glial-derived tumor formation. It would be important to include a wild-type or uninjected control for the pERK and pAKT staining shown in Fig1 I-K to aid in the interpretation of these results. Likewise, quantification of the pERK and pAKT staining would be useful to demonstrate the increase over WT, and would also serve to facilitate comparison with the similar staining in the KPG model (Supp Fig 2D).”

We have updated Fig 1 and Supplementary Fig 3D (formerly Fig 2D), to include histology from tumor-free uninjected control animals, as well as quantifications of p-ERK and p-AKT staining to highlight increased MAPK/AKT signaling pathway activation in our tumor model.

“The authors use a transplantation assay to further test the tumorigenic potential of dissociated cells from glial-derived tumors. Listing the percentage of transplants that generate fluorescent tumor would be helpful to fully interpret these data. Additionally, it was not clear based on the description in the results section that the transplantation assay was an “experimental surrogate” to model the relapse potential of the tumor cell. This is first mentioned in the discussion. The authors may consider adding a sentence for clarity earlier in the manuscript as it helps the reader better understand the logic of the assay.”

We have clarified in the text the percentage of transplants that generated fluorescent tumor (1625%, n=3 independent screens). This is also represented in Fig 5C,D. We also added text when introducing the transplantation assay, explaining that transplantation is frequently used as an experimental surrogate to assess relapse potential, and that our objective was to assess tumor cell propagation in the context of specific manipulations within the TME.

“The authors nicely show high levels of immune cell infiltration and associations between microglia/macrophages and tumor cells. However, a quantification of the emergence of macrophages over time in relation to tumor initiation and growth would provide significant support to the observations of tumor suppressive activity of the phagocytes. Along these lines, the inclusion of a statement about when leukocytes emerge during normal development would be informative for those not familiar with the zebrafish model.”

In zebrafish, microglia colonize the neural retina by 48 hpf, and the optic tectum by 84 hpf (Herbomel et al., 2001), prior to when we typically observe lesions in our p53EPS brains. To validate the emergence of microglia prior to tumor formation in p53EPS, we have now used live confocal imaging through the brains of uninjected control and p53EPS injected zebrafish at 5, 7 and 9 dpf. As expected, microglia were present throughout the cephalic region and in the brain at 5 dpf (120 hpf). At this stage, p53EPS injected zebrafish brains displayed mosaic cellular expression of *her4.1*:mScarlet; however, cells were sparse and diffuse, and no large intensely fluorescent tumor-like clusters were detected at this stage (n=12/12 tumor negative). At 7 dpf, microglia were observed in the brains of control and p53EPS zebrafish; however, at this stage we detected clusters of *her4.1*:mScarlet+ cells (n=5/9), indicative of tumor formation. Lesions were found to be surrounded and/or infiltrated by _mpeg:_EGFP+ microglia. Finally, at 9 dpf *her4.1*:mScarlet+ expression became highly specific to tumor lesions, and these lesions were associated with _mpeg:_EGFP+ microglia/macrophages (n=8/8 of tumor-positive zebrafish). These descriptions along with representative images has been added to Figure 3.

“From the data provided in Figure 4G and Supp Fig 7b, the authors suggest that “increased p53EPS tumor initiation following Irf gene knock-down is a consequence of irf7 and irf8 loss-of-function in the TME.” Given the importance of the local microenvironment highlighted in this study, spatial information on the form of in situ hybridization to identify the relevant location of the expression change would be important to support this conclusion.”

We performed fluorescent in situ hybridization (using HCR RNA-FISH, Molecular Instruments) on whole mount control and *irf7* CRISPR-injected p53EPG animals (*her4.1:***E**GFRvIII +*her4.1:***P**I3KCAH1047R + *her4.1:***G**FP, GFP was used in this case because of probe availability).

Representative confocal projections through tumors, as well as single optical sections are presented and discussed in Figure 4, highlighting the location of *irf7* expression change following gene knock-down. We found significant *irf7* signal in and surrounding p53EPS tumors at early stages of tumor formation_._ This expression was reduced and/or lost following *irf7* CRISPR gene targeting, consistent with RT-PCR data (Supplementary Fig 7).

“The authors used neutral red staining that labels lysosomal-rich phagocytes to assess enrichment at the early stages of tumor initiation. The images in Figure 3 panel A should be labeled to denote the uninjected controls to aid in the interpretation of the data. In Supplemental Figure 6, the neutral red staining in the irf8 CRISPR-injected larvae looks to be increased, counter to the quantification. Can the authors comment if the image is perhaps not representative?”

We have updated Figure 3 and Supplementary Figure 6 to aid in the interpretation of our results. In Fig 3A, we used tumor-negative controls from our injected cohorts. This was done to control for exogenous transgene presence and/or over-expression prior to (or in the absence of) malignant transformation. In Supplementary Fig 6, our images are representative, but we have now used unprocessed images with arrowheads to highlight neutral-red positive foci for clarity. In our original manuscript the images contained software generated markers, which could have obscured and/or confused the neutral red staining we were trying the highlight.

**Recommendations For the Authors:**

**Reviewer #1:**
“The PI 3-kinase does a lot more than just activating mTOR and Akt – I would suggest modifying that sentence in the introduction.”

We have adjusted text in the introduction to reflect the broad role for PI3K signaling.

**Reviewer #2:**
“In Supplemental Fig 1, it would be helpful for the authors to provide a co-stain, such as DAPI to label all nuclei, which would allow the reader to assess the morphology of the cells in the context of the surrounding tissue.”

We have included brightfield images in Supplementary Fig 1, that together with *her4.1*:mScarlet fluorescence, should help readers assess tumor location and morphology in the context of surrounding tissue. Tumor cell morphology at high-resolution can be visualized in Fig 3, Movie 1 and Movie 2.

“The authors state that oncogenic MAPK/AKT pathway activation drives glial-derived tumor formation. The authors may consider testing if the addition of an inhibitor of MAPK signaling may prevent or decrease the formation of glial-derived tumors in this context to further support their results.”

To further assess the role for MAPK activation, we decided to test the effect of 50uM AZD6244 MAPK inhibitor following transplantation of dissociated primary p53EPS cells into syngeneic CG1 strain zebrafish embryos, similar to as previously described (Modzelewska et al., 2016). Following 5 days of drug treatments, we did not detect significant differences in tumor engraftment or in tumor size between DMSO control and AZD6244-treated cohorts, suggesting that MAPK inhibition is not sufficient to prevent p53EPS engraftment and growth in our model. In the future, assessments of on-target drug effects, possible resistance mechanisms, and/or testing MAPK inhibitors in combination with other targeted agents including Akt and/or mTOR inhibitors (Edwards et al., 2006; McNeill et al., 2017; Schreck et al., 2020) will enhance our understanding of potential therapeutic strategies.

**Author response image 2. sa3fig2:** Dorsal views of 8 dpf zebrafish larvae engrafted with her4.1:mScarlet+ p53EPS tumor cells following treatment from 3-8dpf with 0.

“Have the authors tested if EGFR and PI3KCA driven by other neural promoters produce similar results, or not? This would help support the specificity of her4.1 neural progenitors and glia as the cell of origin in this model.”

At this time, we have not tested other neural promoters. However, previous reports describe a zebrafish *zic4*-driven glioblastoma model with mesenchymal-like gene expression (Mayrhofer et al., 2017), supporting neural progenitors as a cell of origin. In the future it will be interesting to test *sox2, nestin,* and *gfap* promoters to further define and support *her4.1*-expressing neural progenitors and glia as the cell of origin in our model.

“Other leukocyte populations, such as neutrophils, can also respond to inflammatory cues. Can the authors comment if neutrophils are also observed in the TME?”

We performed initial assessments of neutrophils in the TME using our expression datasets as well as *her4.1*:EGFRvIII + *her4.1*:PI3KCAH1047R co-injection into Tg(*mpx*:EGFP) strain zebrafish. We observed tumor formation without significant infiltration of *mpx*:EGFP+ neutrophils. Future investigations will be important to assess differences in the contributions of different myeloidderived lineages in the TME of p53EPS, as well as how heterogeneity may be altered depending on different oncogenic drivers and/or stage of tumor progression, as seen in human glioblastoma (Friedmann-Morvinski and Hambardzumyan, 2023). We have added text in the disscussion section of our manuscript to indicate the possibility of neutrophils and/or other immune cell types contributing to p53EPS tumor biology.

**Author response image 3. sa3fig3:** Control-injected tumornegative and tumor-positive Tg(mpx:EGFP) zebrafish at 10 dpf.

“It is not clear if the transcriptomics data has been deposited in a publicly available database, such as the Gene Expression Omnibus (GEO). Sharing of these data would be a benefit to the field and facilitate use in other studies.”

We have uploaded all transcriptomic data to GEO under accession GSE246295.